

Atmospheric
Chemistry
and Physics



# Measurement report: The effect of aerosol chemical composition on light scattering due to the hygroscopic swelling effect

**Rongmin Ren[1], Zhanqing Li[2], Peng Yan[3], Yuying Wang[4], Hao Wu[5], Maureen Cribb[2], Wei Wang[1], Xiao'ai Jin[1], Yanan Li[3], and Dongmei Zhang[1]**

[1]State Key Laboratory of Remote Sensing Science, College of Global Change and Earth System Science, Beijing Normal University, Beijing 100875, China
[2]Department of Atmospheric and Oceanic Science, Earth System Science Interdisciplinary Center, CE1 University of Maryland, College Park, College Park, MD, USA
[3]CMA Meteorological Observation Center, Centre for Atmosphere Watch and Services, CE2 Beijing 100081, China
[4]Key Laboratory for Aerosol–Cloud–Precipitation CE3 of China Meteorological Administration, School of Atmospheric Physics, Nanjing University of Information Science and Technology, Nanjing 210044, China
[5]School of Electrical Engineering, Chengdu University of Information Technology, Chengdu 610225, China

**Correspondence:** Zhanqing Li (zli@atmos.umd.edu)

**Abstract.** TS1 Liquid CE4 water in aerosol particles has a significant effect on their optical properties, especially on light scattering, whose dependence on chemical composition is investigated here using measurements made in southern Beijing in 2019. The effect is measured by the particle light scattering enhancement $f(\mathrm{RH})$, where RH denotes the relative humidity, which is found to be positively and negatively impacted by the proportions of inorganic and organic matter, respectively. Black carbon is also negatively correlated. The positive impact is more robust when the inorganic matter mass fraction was smaller than 40 % ($R = 0.93$, $R$: the Pearson's correlation coefficient), becoming weaker as the inorganic matter mass fraction gets larger ($R = 0.48$). A similar pattern was also found for the negative impact of the organic matter mass fraction. Nitrate played a more significant role in aerosol hygroscopicity than sulfate in Beijing. However, the deliquescence point of ambient aerosols was at about RH = 80 % when the ratio of the sulfate mass concentration to the nitrate mass concentration of the aerosol was high (mostly higher than ∼ 4). Two schemes to parameterize $f(\mathrm{RH})$ were developed to account for the deliquescent and non-deliquescent effects. Using only one $f(\mathrm{RH})$ parameterization scheme to fit all $f(\mathrm{RH})$ processes incurs large errors. A piecewise parameterization scheme is proposed, which can better describe deliquescence and reduces uncertainties in simulating aerosol hygroscopicity.

## 1 Introduction

Atmospheric aerosols have impacts on visibility, the earth–atmosphere radiation budget, clouds, and precipitation via direct and indirect effects (IPCC, 2013). Both effects are associated with the hygroscopic properties of aerosols and relative humidity (RH) of the atmosphere. The particle light scattering enhancement factor, $f(\mathrm{RH}, \lambda)$, is the ratio of the scattering coefficient at an elevated RH level to that under a fixed low RH level (usually RH < 40 %) at a certain light wavelength ($\lambda$). It has been characterized during international field experiments (Fierz-Schmidhauser et al., 2010a, b; Zieger et al., 2010, 2014) and in particular China (Yan et al., 2009; Zhang et al., 2015; Kuang et al., 2016; L. Liu et al., 2018; C. Zhao et al., 2019; Zhao, 2019; Wu et al., 2020).

Aerosol chemical composition has a strong impact on aerosol hygroscopicity (Fierz-Schmidhauser et al., 2010a, b; Wang et al., 2017, 2018, 2019). Zhang et al. (2015) studied the relationship between the scattering enhancement factor and chemical composition in Lin'an, China, finding that nitrate has a stronger effect on aerosol hygroscopicity than

sulfate has, which is partially due to the rigid control of sulfur dioxide ($SO_2$) that reduces the amount of sulfate and increases the content of nitrite (Morgan et al., 2010). Apart from sea salt emissions and gypsum dust emissions during construction containing sulfate, sulfate is mainly formed by the oxidation of its gaseous precursor, $SO_2$, in the atmosphere. In recent years, $SO_2$ emissions have been reduced substantially through a series of effective measures taken in China, like controlling the burning of loose coal and desulfurizing industrial equipment (Zhang et al., 2019). Reducing $SO_2$ in the atmosphere thus directly affects the reduction in the sulfate content of aerosols. The saturated vapor pressure of nitric acid ($HNO_3$) is higher than that of sulfuric acid ($H_2SO_4$), so the availability of ammonia ($NH_3$) is key to the partitioning of $HNO_3$. $HNO_3$ is often neutralized by $NH_3$ after $H_2SO_4$. Therefore, a reduction in $SO_2$ means that more $NH_3$ can be used to neutralize $HNO_3$, leading to higher nitrate concentrations, such as ammonium nitrate ($HN_4NO_3$), in aerosols (Monks et al., 2009). Zieger et al. (2014) analyzed the correlation between the chemical composition of aerosols and $f(RH = 85\%, 550\,nm)$ in Melpitz, Germany, and noted negative and positive impacts by organic and black carbon and by inorganic substances, such as ammonium, respectively. However, the correlation between the mass fractions of individual nitrate ($NO_3^-$) and sulfate ($SO_4^{2-}$) ions and $f(RH = 85\%, 550\,nm)$ were weak. Zieger et al. (2015) also found that the mass fraction of sulfates was strongly correlated with scattering enhancement, while the mass fraction of nitrates had a low correlation in Hyytiälä, Finland. Jin et al. (2020) reported that apart from inorganic matter, e.g., $SO_4^{2-}$ and $NO_3^-$, organic species also significantly contributed to the aerosol liquid water content. They proposed that in the initial phase of a pollution event, the aerosol liquid water content contributed by organic matter accelerated aqueous-phase reactions, converting gaseous precursors into secondary aerosols which then absorb more liquid water.

Air pollution has been and continues to be a serious problem in China, especially in megacities like Beijing. A high scattering enhancement factor is one of the most important factors causing degradation in visibility. Some observational studies of the light hygroscopicity enhancement factor in Beijing have been conducted (Liu et al., 2013; Yang et al., 2015; Zhao et al., 2018; P. Zhao et al., 2019). However, studies exploring the relationship between aerosol chemical composition and the particle light scattering enhancement factor are lacking. Moreover, although several optimal expressions of $f(RH, \lambda)$ for different seasons have been developed (Pan et al., 2009; Yan et al., 2009; Kuang et al., 2016; Wu et al., 2017; Yu et al., 2018; P. Zhao et al., 2019), parameterization of the deliquescent curve of $SO_4^{2-}$ in ambient aerosols has not yet been done for the Beijing–Tianjin–Hebei (BTH) metropolitan region, where aerosol deliquescent phenomena frequently occur (Kuang et al., 2016). It is thus important to develop an optimal parameterization to describe

this deliquescent phenomenon to improve model simulations of aerosol hygroscopicity.

In this study, $f(RH, \lambda)$ at three wavelengths for RH ranging from 40 % to about 90 % was measured by a high-resolution humidified nephelometer system deployed in the southern suburban area of Beijing, China. Other aerosol chemical and physical properties were also simultaneously measured. Humidograms were classified into two categories, i.e., deliquescent and non-deliquescent, each parameterized separately. The parameterization results of deliquescent processes agreed well with observations. This result is useful for improving simulations of $f(RH, \lambda)$ of ambient aerosols during deliquescence in the BTH metropolitan region.

The paper is organized as follows. Section 2 describes the instruments and methods. Section 3 presents and discusses the results of this study, and Sect. 4 provides a summary.

## 2 Instruments and methods

### 2.1 Observation site

A comprehensive field experiment was conducted at the climate observatory of the China Meteorological Administration, located in Yizhuang, Beijing, near the southern Fifth Ring Beltway (39.81° N, 116.48° E) surrounded primarily by residential communities and industrial parks (Fig. S1 in the Supplement). Measurements made here can characterize the aerosol chemical and physical properties of a typical suburban area of this megacity in the North China Plain. Equipped with a multitude of instruments measuring, for example, optical, hygroscopic, and chemical properties of aerosols (Z. Li et al., 2019), this study employs only those measurements acquired from 19 September to 4 October 2019. The instruments used in this field experiment include a dual-nephelometer system (Aurora 3000, Ecotech), an aerosol chemical speciation monitor (ACSM; Aerodyne Research Inc.), and a seven-wavelength aethalometer (AE33, Magee Scientific). They were all located in a mobility container on the ground. There are two air conditioners inside the container whose temperature was maintained at about 23°C. Sample air (16.7 L min$^{-1}$) went through a PM$_{2.5}$ cyclone inlet at about 4 m above the ground, which only allowed particles with an aerodynamic diameter smaller than 2.5 μm to enter, and was then dried by a Nafion dryer (MD-700-36F-3, Perma Pure LLC). The average RH within the sampling line was about 30 %. The sample air was not heated.

### 2.2 Instruments

A dual-nephelometer system with a high time resolution was used to measure the particle light scattering enhancement factor of aerosol. After an aerosol sample passed through the Nafion dryer, the dry sample flow (RH < 40 %) was divided into two routes. One sample flow (5 L min$^{-1}$) went directly into the dry nephelometer. The other sample flow (5 L min$^{-1}$)

passed through an annular concentric humidifying tube (MD-700-6F-3, Perma Pure LLC). The water vapor controlled by the temperature of the liquid water in the outer annulus of the tube passes through a Nafion membrane, humidifying aerosols in the inner tube (Carrico et al., 1998). The temperature of the liquid water was controlled by adjusting the power of the water baths. The sample flow was then humidified to a given RH and channeled into the wet nephelometer. The scattering coefficients under dry (the mean $\pm$ standard deviation value of RH was $28.75 \pm 5.50\%$) and wet ambient conditions were measured synchronously by the two nephelometers (Yan et al., 2009). To improve the performance of this system and to decrease the amount of time needed to undergo one aerosol humidifying process, two water baths were used in turn to heat the water circulating in the interior layer of the humidifying tube (Liu and Zhao, 2016).

Since the RH of aerosols inside the nephelometers was constantly changing and real measured data at every moment were needed, the nephelometers operated without Kalman filters. Manual "full calibration" and zero check and span check of the two nephelometers were performed at 10:30 (all times in Beijing time, UTC + 8 h) CE5 on 19 September 2019. The calibration tolerance of the zero check was $\pm 2\,\mathrm{Mm}^{-1}$, and that of the span check was $\pm 2\%$ of the span point. Calibrations of the two nephelometers in the dry state were consistent (Fig. S2). The truncation and illumination correction of the scattering coefficients has been done following Müller et al. (2011), which was developed specifically for Ecotech nephelometers originating from Anderson and Ogren (1998) for TSI nephelometers. For the $f(\mathrm{RH})$ calculations, there is no truncation and illumination correction applied to the scattering coefficients of both dry and humidified types of nephelometer. The comparison of the deviation between corrected and uncorrected $f(\mathrm{RH} = 85\%, 525\,\mathrm{nm})$ is shown in Fig. S3. The linear least square regression slop $\pm$ standard deviation is $1.064 \pm 0.002$, the intercept $\pm$ standard deviation is $-0.082 \pm 0.004$, and $R$ is 0.999. The fitted line is very close to the line of $1:1$. The uncorrected $f(\mathrm{RH} = 85\%, 525\,\mathrm{nm})$ is a little lower than the corrected $f(\mathrm{RH} = 85\%, 525\,\mathrm{nm})$.

Because the RH levels measured by the probe built into the optical chamber of the wet nephelometer ($\mathrm{RH}_{\mathrm{chamber}}$) were imprecise, the $\mathrm{RH}_{\mathrm{chamber}}$ was corrected in this paper. First, a set of calibrated RH and temperature probes was placed at the inlet of the wet nephelometer, and another set was placed at the outlet of the wet nephelometer, obtaining 1 min averages of RH and temperature. We used Vaisala HMP110 probes with accuracies of $\pm 0.2\,°\mathrm{C}$ for the 0–40 °C temperature range and $\pm 1.5\%$ RH and $\pm 2.5\%$ RH for the 0 % RH–90 % RH and 90 % RH–100 % RH TS2 ranges, respectively. The temperatures measured by these three probes were different. However, in principle, the dew point temperatures ($T_{\mathrm{dew\text{-}point}}$) at these three positions are all the same. Since the RH and temperature probes at the outlet of the wet nephelometer ($\mathrm{RH}_{\mathrm{outlet}}$ and $T_{\mathrm{outlet}}$) were less affected by the hu-

midifier, $\mathrm{RH}_{\mathrm{outlet}}$ and $T_{\mathrm{outlet}}$ were used to calculate $T_{\mathrm{dew\text{-}point}}$ at this position using Eq. (1) (Wanielista et al., 1997; James et al., 2015):

$$T_{\mathrm{dew\text{-}point}} = \mathrm{RH}_{\mathrm{outlet}}^{\frac{1}{8}}(112 + 0.9T_{\mathrm{outlet}}) + 0.1T_{\mathrm{outlet}} - 112. \quad (1)$$

We assume that $T_{\mathrm{dew\text{-}point}}$ was approximately the same as that in the optical chamber of the wet nephelometer. Based on the temperature in the optical chamber ($T_{\mathrm{chamber}}$) and $T_{\mathrm{dew\text{-}point}}$, the actual RH in the optical chamber ($\mathrm{RH}_{\mathrm{chamber}}$) can be calculated by rearranging Eq. (1), i.e.,

$$\mathrm{RH}_{\mathrm{chamber}} = \left( \frac{112 - 0.1T_{\mathrm{chamber}} + T_{\mathrm{dew\text{-}point}}}{112 + 0.9T_{\mathrm{chamber}}} \right). \quad (2)$$

The dual-nephelometer system with a high time resolution in this study was calibrated with ammonium sulfate, $(\mathrm{NH_4})_2\mathrm{SO_4}$, whose deliquescence RH (DRH) was 80 % at 298 K (Cheung et al., 2015). Figure S4 shows that the measured phase transition occurs at RH = 80.07 %. It illustrates that the RH inside the nephelometer chamber is correct and that the system is functioning properly.

An ACSM measured the mass concentrations of non-refractory aerosol chemical species, including $\mathrm{SO_4^{2-}}$, $\mathrm{NO_3^-}$, ammonium ($\mathrm{NH_4^+}$), chlorine (Chl), and organics (Orgs) in particulate matter with diameters less than 2.5 µm ($\mathrm{PM}_{2.5}$). The mass concentration of the equivalent black carbon (eBC) could be retrieved from the measurements of AE33. The Chinese Ministry of Ecology and Environment network and the Beijing Municipal Environmental Monitoring Center (http://106.37.208.233:20035/ TS3 and http://www.bjmemc.com.cn/ TS4) provided mass concentrations of $\mathrm{PM}_{2.5}$ measured at the Yizhuang station, about 3 km southeast of the observatory. The LI-COR eddy covariance system (this system includes WindMaster Pro, LI-7500A, and Smart2-00171, LI-COR) measured various meteorological parameters.

## 2.3 Methods

The particle light scattering enhancement factor, $f(\mathrm{RH}, \lambda)$, is defined as

$$f(\mathrm{RH}, \lambda) = \frac{\sigma_{\mathrm{sp}}(\mathrm{RH}, \lambda)}{\sigma_{\mathrm{sp}}(\mathrm{RH}_{\mathrm{dry}}, \lambda)}, \quad (3)$$

where $\sigma_{\mathrm{sp}}(\mathrm{RH}, \lambda)$ represents the scattering coefficient at an elevated RH (usually RH > 40 %), and $\sigma_{\mathrm{sp}}(\mathrm{RH}_{\mathrm{dry}}, \lambda)$ is the scattering coefficient in the dry state (usually RH < 40 %) at wavelength $\lambda$. Values of $f(\mathrm{RH}, \lambda)$ are generally greater than 1 and increase with increasing RH. In this study, we assume that the aerosol is in the dry state when RH < 40 %. This means that in theory, $f(\mathrm{RH})$ should equal 1 when RH is lower than 40 %. However, due to systematic errors and differences in RH measured synchronously by the dry nephelometer and the wet nephelometer, the measured $f(\mathrm{RH} < 40\%)$ has small fluctuations and does not equal 1.

Therefore, $f(\mathrm{RH} > 40\,\%)$ was normalized as

$$f(\mathrm{RH} > 40\,\%)_{\mathrm{normalized}} = \left(\frac{f(\mathrm{RH} > 40\,\%)}{f(\mathrm{RH} < 40\,\%)_{\mathrm{averaged}}}\right). \quad (4)$$

Here, $f(\mathrm{RH} < 40\,\%)_{\mathrm{averaged}}$ is the corrected coefficient averaged over the whole dataset of RH < 40 %.

The absorption coefficient of PM$_{2.5}$ at 520 nm was calculated by the eBC monitor (Han et al., 2015; Zou et al., 2019). To facilitate comparisons, absorption coefficients at 520 nm were transformed into those at 525 nm by assuming that the absorption coefficient is inversely proportional to the wavelength (Bond and Bergstrom, 2006; C. Liu et al., 2018). The quantity $\omega_{0(525\,\mathrm{nm})}$ is the aerosol single-scattering albedo at 525 nm. The dependence of light scattering on wavelength is described by the Ångström exponent ($\alpha_{(\lambda_1-\lambda_2)}$), an index describing the particle size:

$$\alpha_{(\lambda_1-\lambda_2)} = \frac{\log\sigma_{\mathrm{sp}}(\lambda_1) - \log\sigma_{\mathrm{sp}}(\lambda_2)}{\log\lambda_2 - \log\lambda_1}. \quad (5)$$

Here, $\alpha_{(450-635\,\mathrm{nm})}$ between 450 and 525 nm was calculated using Eq. (5).

The following parameter ($F_{\mathrm{org}}$) denotes the relative amount of organic and inorganic matter:

$$F_{\mathrm{org}} = \frac{C_{\mathrm{c}}}{C_{\mathrm{c}} + C_{\mathrm{i}}}, \quad (6)$$

where $C_{\mathrm{c}}$ is the organic matter mass concentration measured by the ACSM, and $C_{\mathrm{i}}$ represents the mass concentration of inorganic salts like $(\mathrm{NH_4})_2\mathrm{SO_4}$, ammonium bisulfate ($\mathrm{NH_4HSO_4}$), and ammonium nitrate ($\mathrm{NH_4NO_3}$).

Deliquescence of ambient aerosols was present throughout the study period. To identify this process and describe its magnitude in the 78 % RH–82 % RH range, the hysteresis index $\eta$ is defined as (Zieger et al., 2010)

$$\eta = 1 - \frac{\gamma_{<78\,\%}}{\gamma_{>82\,\%}}. \quad (7)$$

The terms $\gamma_{<78\,\%}$ and $\gamma_{>82\,\%}$ are the fit parameters of the $f(\mathrm{RH})$ parametrization scheme,

$$f(\mathrm{RH}) = (1 - \mathrm{RH})^{-\gamma}, \quad (8)$$

at RH < 78 % and RH > 82 %, respectively. The parameter $\gamma$ is retrieved from Eq. (8) using the whole RH range. It can replace $f(\mathrm{RH})$ in a wider RH range (Doherty et al., 2005; Quinn et al., 2005; Zhang et al., 2015). The theoretical range of $\eta$ is 0 to 1. The $\gamma_{<78\,\%}$ and $\gamma_{>82\,\%}$ terms, respectively, represent the magnitudes of the scattering enhancement when RH < 78 % and RH > 82 %. If the values of $\gamma_{<78\,\%}$ and $\gamma_{>82\,\%}$ are about the same, then $\eta$ will be close to 0. This suggests that $f(\mathrm{RH})$ increases slowly and continuously when 78 % < RH < 82 %, and no deliquescence is found. However, when the value of $\gamma_{>82\,\%}$ is much higher than $\gamma_{<78\,\%}$, $\eta$ approaches 1. This explains why the $f(\mathrm{RH})$ cycle has a jump at 78 % < RH < 82 %, i.e., very distinct deliquescence occurring in the RH range of 78 % to 82 %. Here, when $\eta$ is higher than 0.4, deliquescence occurs.

## 3 Results and discussion

### 3.1 Overview

Figure 1 depicts the hourly averaged time series of the light-scattering coefficient ($\sigma_{\mathrm{sp},525\,\mathrm{nm}}$), the absorption coefficient ($\sigma_{\mathrm{ap},525\,\mathrm{nm}}$), the single-scattering albedo ($\omega_{0(525\,\mathrm{nm})}$), the scattering Ångström exponent ($\alpha_{(450-635\,\mathrm{nm})}$), and the particle light scattering enhancement factor at RH = 85 %, $f(\mathrm{RH} = 85\,\%,\ 525\,\mathrm{nm})$, at the main observatory, and the mass concentration of PM$_{2.5}$ measured at the Yizhuang station from 19 September to 4 October 2019. During this period, the hourly averaged $\sigma_{\mathrm{sp},525\,\mathrm{nm}}$ ranged from 3 to 799 Mm$^{-1}$ (Fig. 1a), with a mean ± standard deviation value of 245 ± 168 Mm$^{-1}$ (Table 1). The hourly averaged $\sigma_{\mathrm{ap},525\,\mathrm{nm}}$ varied from 7 to 135 Mm$^{-1}$ (Fig. 1b), with a mean ± standard deviation value of 50 ± 24 Mm$^{-1}$ (Table 1). Figure 1c and f show that $\omega_{0(525\,\mathrm{nm})}$ increased as the PM$_{2.5}$ concentration increased. The hourly averaged $\omega_{0(525\,\mathrm{nm})}$ during the observation period ranged from 0.24 to 0.98, with an overall mean ± standard deviation value of 0.77 ± 0.15. The mean ± standard deviation values of $\omega_{0(525\,\mathrm{nm})}$ during clean (PM$_{2.5} \leq 35\,\mu\mathrm{g\,m}^{-3}$), moderately polluted (35 $\mu\mathrm{g\,m}^{-3}$ < PM$_{2.5} \leq 75\,\mu\mathrm{g\,m}^{-3}$), and heavily polluted (PM$_{2.5} > 75\,\mu\mathrm{g\,m}^{-3}$) periods were 0.65 ± 0.16, 0.83 ± 0.08, and 0.93 ± 0.04, respectively (Table 1).

The $\omega_{0(525\,\mathrm{nm})}$ increased as PM$_{2.5}$ pollution increased, indicating that during the contamination process, the proportion of aerosol components with strong scattering properties increased, and the proportion of aerosol components with strong absorbing properties decreased. Higher values of $\omega_{0(525\,\mathrm{nm})}$ usually occurred when the wind was from the northeast at a speed of 1–2 m s$^{-1}$ and when relatively stronger winds were from the southeast (Fig. 2a), which was frequently accompanied by a high proportion of inorganic matter (Fig. 2e) and low proportions of eBC (Fig. 2d) and organic matter (Fig. 2f). Figure 1d shows that the range of $\alpha_{(450-635\,\mathrm{nm})}$ was narrow at most times during the observation period. Relatively larger particles with lower values of $\alpha_{(450-635\,\mathrm{nm})}$ generally occurred when weak winds were from the east and southeast (Fig. 2b).

During the observation period, $f(\mathrm{RH} = 85\,\%,\ 525\,\mathrm{nm})$ ranged from 1.15 to 1.86, meaning a 1.15- to 1.86-fold increase in the scattering coefficient at RH = 85 % compared to dry conditions (Fig. 1e). Daily average values of $f(\mathrm{RH} = 85\,\%,\ 525\,\mathrm{nm})$ varied between 1.32 and 1.74, with low values (< 1.40) on 19 and 24 September and relatively high values (> 1.70) on 22 and 28 September and 2 October. Organic matter mass fractions were larger than 52 %, and inorganic matter mass fractions were generally smaller when $f(\mathrm{RH} = 85\,\%,\ 525\,\mathrm{nm})$ was less than 1.40. However, high values of $f(\mathrm{RH} = 85\,\%,\ 525\,\mathrm{nm})$ in this study were often closely correlated with large fractions of water-soluble ions, such as NO$_3^-$ and SO$_4^{2-}$, in PM$_{2.5}$. Inorganic matter mass fractions were larger than 53 %, and or-

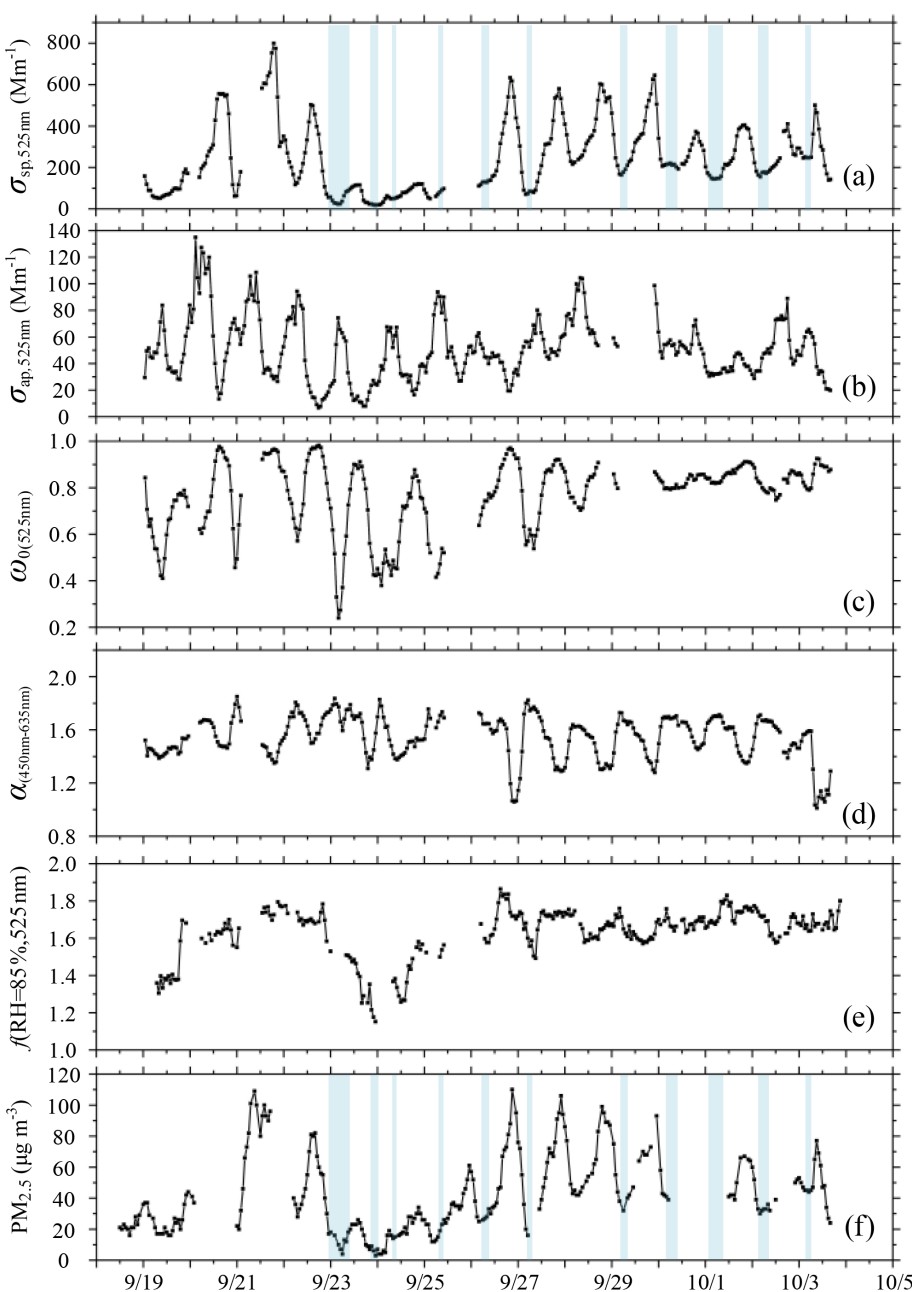

**Figure 1.** Hourly averaged time series (18 September to 4 October 2019) of the **(a)** scattering coefficient ($\sigma_{sp,525\,nm}$) measured by the dry nephelometer (unit: $Mm^{-1}$), **(b)** absorption coefficient ($\sigma_{ap,525\,nm}$; unit: $Mm^{-1}$), **(c)** single-scattering albedo ($\omega_{0(525\,nm)}$), **(d)** scattering Ångström exponent ($\alpha_{(450-635\,nm)}$), **(e)** particle light scattering enhancement factor at RH = 85 % ($f(RH = 85\,\%, 525\,nm)$), and **(f)** mass concentration of $PM_{2.5}$ (unit: $\mu g\,m^{-3}$) measured at the Yizhuang station. The segments of the time series with a blue background represent the occurrence of deliquescence. The timescale is Beijing time (UTC + 8 h). The date in this figure is in the month/day format.

ganic matter mass fractions were relatively small when the $f(RH = 85\,\%, 525\,nm)$ was greater than 1.70. The campaign mean $\pm$ standard deviation values of $f(RH = 85\,\%, 525\,nm)$ were $1.64 \pm 0.13$ (Table 1). Figure 2c reveals that strongly hygroscopic aerosols with high values of $f(RH = 85\,\%, 525\,nm)$ primarily came from the southeast sector. The proportion of secondary inorganics with strong hygroscopic

abilities in aerosols from this direction was high, while the proportion of organic matter with weak hygroscopic abilities was low (Fig. 2e–f). Figure 2d indicates that the mass fraction of eBC with weak hygroscopicity was slightly low in the southeast sector when wind speeds were lower than $4\,m\,s^{-1}$. However, when wind speeds were higher than $4\,m\,s^{-1}$, the mass fraction of eBC was relatively high in this direction.

**Table 1.** Average $\sigma_{\text{sp},525\,\text{nm}}$, $\sigma_{\text{ap},525\,\text{nm}}$, $\omega_{0(525\,\text{nm})}$, $\alpha_{(450-635\,\text{nm})}$, $f(\text{RH} = 85\,\%, 525\,\text{nm})$, and $PM_{2.5}$ mass concentration values at different pollution levels.

| | Entire | PM$_{2.5}$ pollution levels (µg m$^{-3}$) | | |
| --- | --- | --- | --- | --- |
| | observation period | Very clean (PM$_{2.5} \leq 35$) | Moderately polluted ($35 < PM_{2.5} \leq 75$) | Heavily polluted (PM$_{2.5} > 75$) |
| $\sigma_{\text{sp},525\,\text{nm}}$ (Mm$^{-1}$) | $245 \pm 168$ | $89 \pm 45$ | $279 \pm 93$ | $530 \pm 75$ |
| $\sigma_{\text{ap},525\,\text{nm}}$ (Mm$^{-1}$) | $50 \pm 24$ | $43 \pm 21$ | $51 \pm 20$ | $50 \pm 27$ |
| $\omega_{0(525\,\text{nm})}$ (–) | $0.77 \pm 0.15$ | $0.65 \pm 0.16$ | $0.83 \pm 0.08$ | $0.93 \pm 0.04$ |
| $\alpha_{(450-635\,\text{nm})}$ (–) | $1.55 \pm 0.16$ | $1.58 \pm 0.15$ | $1.54 \pm 0.15$ | $1.33 \pm 0.13$ |
| $f(\text{RH} = 85\,\%, 525\,\text{nm})$ (–) | $1.64 \pm 0.13$ | $1.49 \pm 0.16$ | $1.70 \pm 0.06$ | $1.71 \pm 0.05$ |
| PM$_{2.5}$ (µg m$^{-3}$) | $44 \pm 25$ | $22 \pm 9$ | $51 \pm 12$ | $90 \pm 9$ |

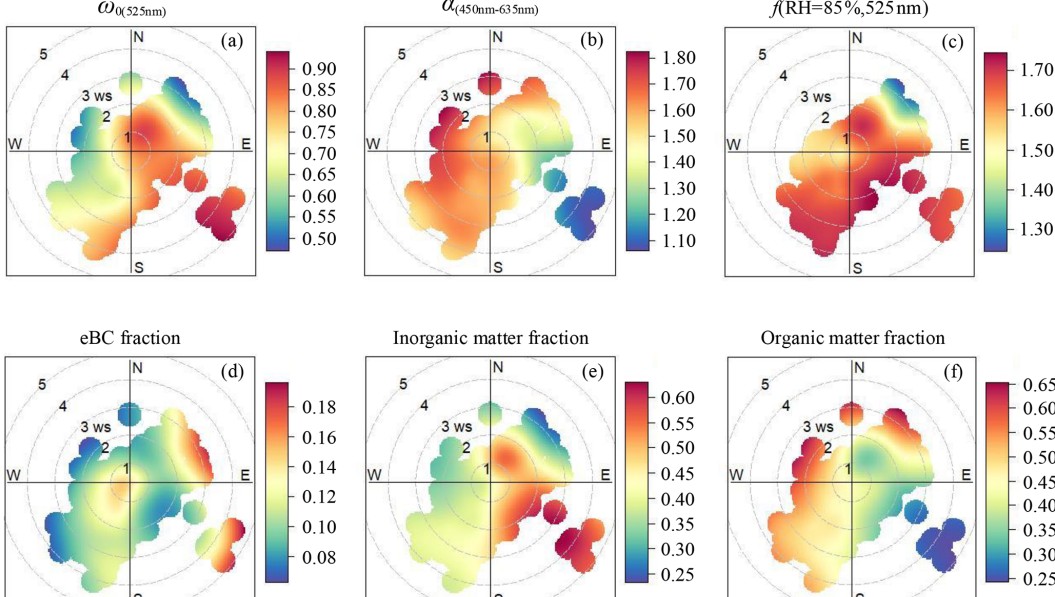

**Figure 2.** Wind dependence of **(a)** the single-scattering albedo ($\omega_{0(525\,\text{nm})}$), **(a)** the scattering Ångström exponent ($\alpha_{(450-635\,\text{mm})}$), **(c)** the particle light scattering enhancement factor at RH = 85 % ($f(\text{RH} = 85\,\%, 525\,\text{nm})$), **(d)** the mass fraction of eBC, **(e)** the mass fraction of inorganic matter, and **(f)** the mass fraction of organic matter. The circular contours show the average change in wind speed and direction.

Of all data associated with southeast winds, only three cases were identified with wind speeds higher than $4\,\text{m s}^{-1}$, likely winds of short duration so not representative. In the northeast direction, high values of $f(\text{RH} = 85\,\%, 525\,\text{nm})$ occurred when the wind speed was lower than $2\,\text{m s}^{-1}$. The hygroscopic capacity of aerosols also weakened as the wind speed increased (Fig. 2c). The proportion of secondary inorganics with strong hygroscopicity decreased with increasing wind speed (Fig. 2e), while the proportion of substances with weak hygroscopicity, such as organic matter and eBC, increased with increasing northeasterly wind speeds (Fig. 2d, f). Furthermore, aerosols from the southwest and southern sectors within the wind-speed range of 2 to $4\,\text{m s}^{-1}$ had higher scattering enhancement factors (Fig. 2c) mainly because of the deliquescence of sulfates in the ambient aerosols. The specific reasons are explained in detail in Sect. 3.3. Figure S5

shows that apart from the lower values (10th percentile values in Table S1), a small wavelength dependence in scattering enhancement factor is found in all other percentiles, with a stronger wavelength dependence for high values of $f(\text{RH} = 85\,\%)$. Zieger et al. (2014) and Zhang et al. (2015) obtained similar results for Melpitz, Germany, and Linan, China, respectively.

Figure S6a to b show the time series of mass concentrations and mass fractions, respectively, of submicron aerosols, i.e., organic matter, nitrate, sulfate, ammonium, chloride, and eBC in PM$_{2.5}$. The rightmost pie chart in Fig. S6c shows that during the entire observation period, organic matter was the major component of PM$_{2.5}$, accounting for 39 %. Nitrate and sulfate comprised similar fractions of PM$_{2.5}$, i.e., 21 % and 19 %, respectively. The mass fraction of nitrate was slightly larger than that of sulfate. Note that eBC ac-

counted for 11 % of PM$_{2.5}$ during the entire measurement period. Two special periods were noted. One started in the afternoon of 21 September and ended late morning on 22 September (Sect. I in Fig. S6a). First, the concentrations of all chemical components during this period were high. As shown by the leftmost pie chart in Fig. S6c, the mass fraction of nitrate was the largest, accounting for 33 % of the total mass fraction on average, lasting a long time. However, compared with the proportion of inorganic matter (66 %), the mass fraction of organic matter was much smaller, accounting for 27 % of the total mass fraction. Accordingly, the $f(\mathrm{RH} = 85\,\%, 525\,\mathrm{nm})$ remained at a high level during this period (Fig. 1e). The other special period was on 24 September (Sect. II in Fig. S6a). The mass concentrations of all aerosol species remained low. The middle pie chart in Fig. S6c clearly demonstrates that organic matter comprised the main fraction of PM$_{2.5}$, accounting for 55 % on average, followed by eBC. The fraction of nitrate was especially small during this clean period. The fraction of sulfate, accounting for 16 %, was 4 times that of nitrate, which was an advantage for deliquescence in this period (discussed in more detail in Sect. 3.3). Although the proportions of hydrophobic organic matter and eBC in aerosols during this period was very high, $f(\mathrm{RH} = 85\,\%, 525\,\mathrm{nm})$ was not the lowest during the whole observation period because the $f(\mathrm{RH} = 85\,\%, 525\,\mathrm{nm})$ of deliquescence was higher than the normal value. Note that PM$_{2.5}$ aerosols at the observatory in suburban Beijing were faintly acidic during the observation period (Fig. S7), benefitting the hygroscopic enhancement of ambient aerosols.

## 3.2 The relationship between the particle light scattering enhancement factor and aerosol chemical composition

Figure 3 displays $f(\mathrm{RH} = 85\,\%, 525\,\mathrm{nm})$ as a function of the main chemical component mass fractions. The total aerosol mass concentration is the sum of mass concentrations of all chemical constituents, including nitrate, sulfate, ammonium, chloride, and organic matter measured by the ACSM and eBC retrieved by the AE33. The mass fractions of individual chemical components were calculated by respectively dividing the mass concentrations of sulfate, nitrate, ammonium, and eBC by the sum of all chemical constituents. The $f(\mathrm{RH} = 85\,\%, 525\,\mathrm{nm})$ and eBC were negatively correlated, with a correlation coefficient $R$ equal to $-0.62$ (Fig. 3d). A positive correlation is seen between $f(\mathrm{RH} = 85\,\%, 525\,\mathrm{nm})$ and the three other inorganic substances' mass fractions because of their hygroscopic characteristics. The ammonium mass fraction had the strongest positive correlation ($R = 0.78$) with $f(\mathrm{RH} = 85\,\%, 525\,\mathrm{nm})$ (Fig. 3c). The reason is that ammonium is the common positive ion of $(\mathrm{NH_4})_2\mathrm{SO_4}$ and $\mathrm{NH_4NO_3}$, two major salts of inorganic substances in aerosols. The relationship between $f(\mathrm{RH} = 85\,\%, 525\,\mathrm{nm})$ and ammonium is thus similar to that between $f(\mathrm{RH} = 85\,\%, 525\,\mathrm{nm})$ and inorganic content. The hygroscopic properties

were different for $\mathrm{NH_4NO_3}$ and $(\mathrm{NH_4})_2\mathrm{SO_4}$. As expected, $f(\mathrm{RH} = 85\,\%, 525\,\mathrm{nm})$ was positively correlated with the sum of the nitrate and sulfate mass fractions (slope $= 1.03$ and $R = 0.79$; Fig. S8), similar to the correlation between $f(\mathrm{RH} = 85\,\%, 525\,\mathrm{nm})$ and the inorganic mass fraction. Wu et al. (2017) and Zieger et al. (2014) reported similar results.

Figure 4 shows $f(\mathrm{RH} = 85\,\%, 525\,\mathrm{nm})$ as a function of the mass fractions of organic and inorganic matter. The mass fractions of inorganic and organic matter were respectively calculated by dividing inorganic matter (the sum of nitrate, sulfate, ammonium, and chloride) and organic matter mass concentrations by the total mass concentration. The inorganic mass fraction was positively correlated with $f(\mathrm{RH} = 85\,\%, 525\,\mathrm{nm})$ because of the high hygroscopicity of the inorganic compounds, while organic substances were negatively correlated with $f(\mathrm{RH} = 85\,\%, 525\,\mathrm{nm})$ because of their lower hygroscopicity (P. Zhao et al., 2019). Both correlation coefficients were similar to those from previous studies (Zieger et al., 2014; Zhang et al., 2015; Wu et al., 2017). Also, the absolute values of both the slopes and corresponding standard deviations found here ($0.80 \pm 0.04$ and $1.00 \pm 0.06$ for $f(\mathrm{RH} = 85\,\%, 525\,\mathrm{nm})$ as a function of inorganic and organic matter mass fractions, respectively) were similar to those reported in Lin'an, China ($0.96 \pm 0.02$ and $1.20 \pm 0.04$, respectively; Zhang et al., 2015), but much lower than those observed in Melpitz, Germany ($2.2 \pm 0.078$ and $3.1 \pm 0.1$, respectively; Zieger et al., 2014). This might be because the $f(\mathrm{RH} = 85\,\%, 525\,\mathrm{nm})$ measured in Melpitz, Germany, was much higher than that in Lin'an and Beijing. Ambient aerosols in Melpitz, Germany, were partially affected by sea salt, like sodium chloride, transported from the North Sea and being highly hygroscopic. Marine aerosols have a higher hygroscopicity than aerosols influenced more by human activity.

Distinguishing between data points below and above the 40 % organic mass fraction level in Fig. 4, the absolute values of the linear regression slope and $R$ for data below 40 % were lower than those for data above 40 %. However, for the inorganic mass fraction (left panels of Fig. 4), the absolute values of the linear regression slope and $R$ for data below 40 % were higher than those for data above 40 %. This indicates that the positive correlation between $f(\mathrm{RH} = 85\,\%, 525\,\mathrm{nm})$ and the inorganic matter mass fraction was very strong when inorganic matter mass fractions were smaller than 40 %. However, when inorganic matter mass fractions were larger than 40 %, the increasing tendency in $f(\mathrm{RH} = 85\,\%, 525\,\mathrm{nm})$ as the proportion of inorganic matter increased slowly decreased. Inversely, the negative correlation between $f(\mathrm{RH} = 85\,\%, 525\,\mathrm{nm})$ and the organic matter mass fraction was slightly weak when organic matter mass fractions were smaller than 40 %. When organic matter mass fractions were larger than 40 %, the decreasing tendency in $f(\mathrm{RH} = 85\,\%, 525\,\mathrm{nm})$ as the proportion of organic matter decreased became more robust. There may be several reasons for this phenomenon. First, particle size may

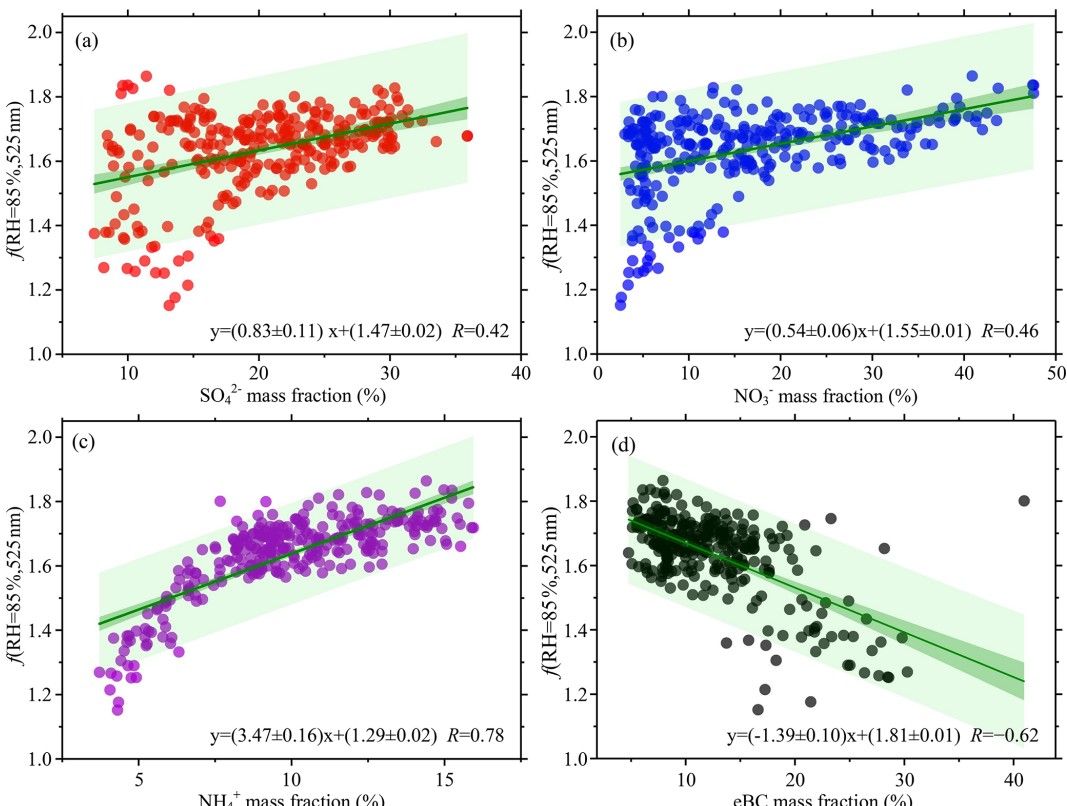

**Figure 3.** The particle light scattering enhancement factor $f(\mathrm{RH} = 85\,\%, 525\,\mathrm{nm})$ as a function of different aerosol chemical component mass fractions measured by the ACSM and the AE33: **(a)** sulfate ($\mathrm{SO_4^{2-}}$) mass fraction, **(b)** nitrate ($\mathrm{NO_3^-}$) mass fraction, **(c)** ammonium ($\mathrm{NH_4^+}$) mass fraction, and **(d)** black carbon (eBC) mass fraction. Solid green lines represent bivariate linear regressions. The dark-green shaded areas denote 95 % confidence levels, and the light-green shaded areas show the 95 % prediction bands for the fits. The linear regression function and the Pearson's correlation coefficient ($R$) are given in each panel.

be one of the most important factors to explain this. For particles with strong hygroscopicity, if their particle number size distribution tends towards large particle sizes, their hygroscopic growth ability may be similar to that of smaller particles with weak hygroscopicity (Zieger et al., 2010, 2013; Wang et al., 2017, 2018). On the one hand, the $f(\mathrm{RH})$ of particles usually decreases with increasing particle size (Zieger et al., 2013), resulting in a lower $f(\mathrm{RH})$ for larger particles. On the other hand, the larger particles' amplification effect of scattering cross section because of hygroscopic growth is weaker than that of smaller particles (Wu et al., 2017). It may be that when there is a high proportion of inorganic matter and a low proportion of organic matter, the inorganic matter is mainly composed of relatively large particles. It is possible that the $f(\mathrm{RH} = 85\,\%, 525\,\mathrm{nm})$ of aerosols with a high level of inorganic matter and a low level of organic matter is not as high as expected due to the compensating effect between the size and the chemical composition of aerosol. Second, the higher mass concentration of ambient aerosols is maybe another reason. Overall, when the mass concentration of inorganic substances was less than 40 %, the total mass concentration of aerosols was relatively low in this experi-

ment. When the proportion of inorganic matter was higher than 40 %, the total mass concentration of aerosols was high, with a clear inhibiting effect of high aerosol mass concentration to the $f(\mathrm{RH} = 85\,\%, 525\,\mathrm{nm})$ (Fig. 4e). Finally, it is also possible that when the proportion of inorganic matter is very high, aerosols absorb too much water vapor, leading to insufficient ambient water vapor.

The green dots in Fig. S9a and b represent deliquescence. The range of inorganic mass fraction in these deliquescent processes was from 30 % to 50 %, and the range of organic mass fraction was from 40 % to 60 %. A comparison between Fig. 4a and c and b and d shows that the proportion of sulfate in the total aerosol was much higher than that of nitrate for these deliquescent processes. Meanwhile, the $f(\mathrm{RH} = 85\,\%, 525\,\mathrm{nm})$ of these deliquescent processes were all above the best-fit regression lines. This demonstrates that the scattering enhancement factor at 85 % RH of deliquescence was generally higher than that of the non-deliquescent process. Kuang et al. (2016) also drew the same conclusion.

Comparing Fig. 4a and c, as the proportion of inorganic salts in the total aerosol increased, the proportion of nitrate in the total aerosol also increased, unlike the proportion of

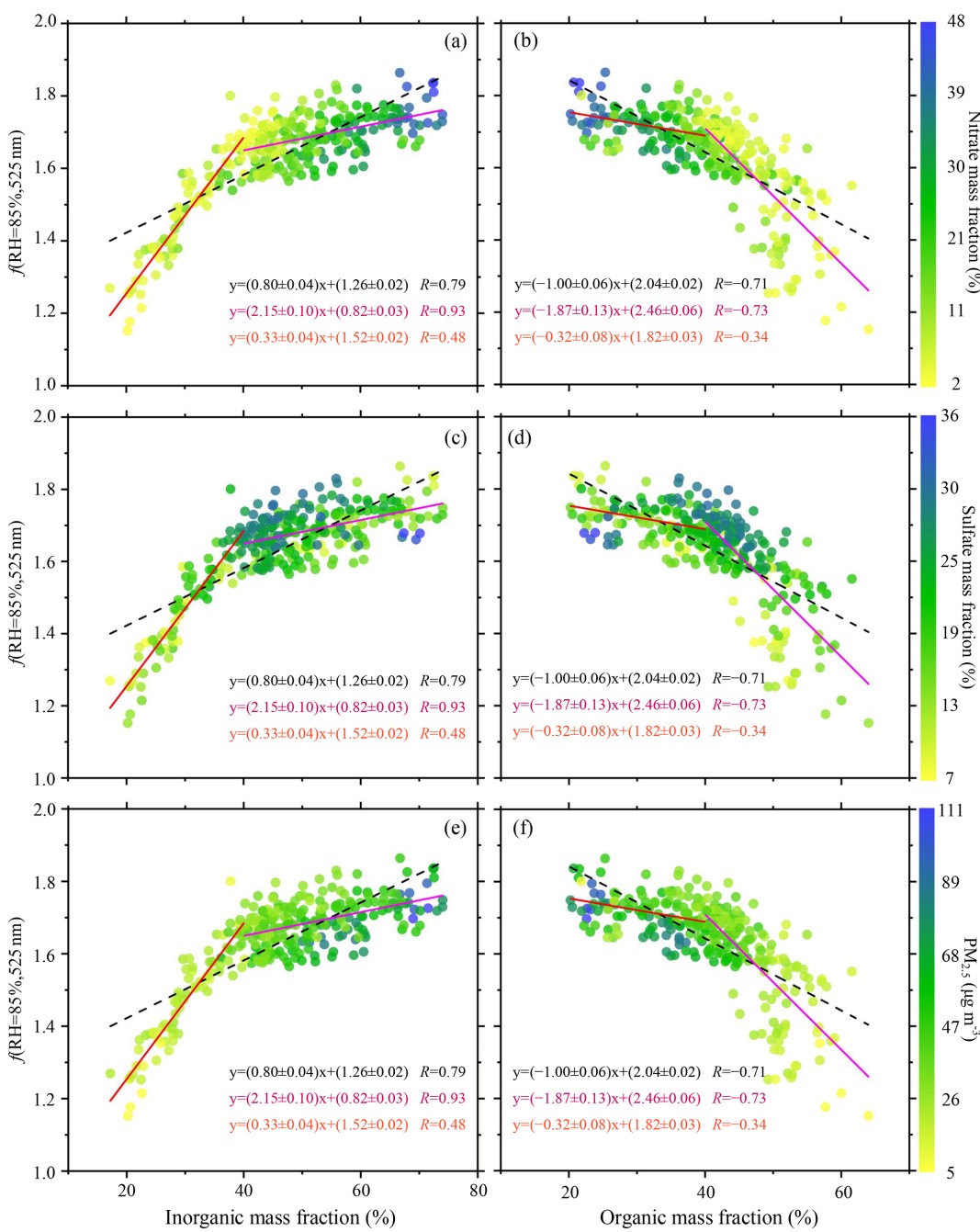

**Figure 4.** The particle light scattering enhancement factor $f$(RH = 85 %, 525 nm) as a function of organic matter mass fraction and inorganic matter mass fraction measured by the ACSM and the AE33: $f$(RH = 85 %, 525 nm) as a function of **(a)** inorganic matter mass fraction and **(b)** organic matter mass fraction colored by the nitrate mass fraction; $f$(RH = 85 %, 525 nm) as a function of **(c)** inorganic matter mass fraction and **(d)** organic matter mass fraction colored by the sulfate mass fraction; and $f$(RH = 85 %, 525 nm) as a function of **(e)** inorganic matter mass fraction and **(f)** organic matter mass fraction colored by the mass concentration of PM$_{2.5}$. Dotted black lines denote bivariate linear regressions. The red and magenta lines are the best-fit linear regression lines through data points associated with mass fractions smaller than 40 % and larger than 40 %, respectively. The linear regression functions and the Pearson's correlation coefficients ($R$) are given in each panel.

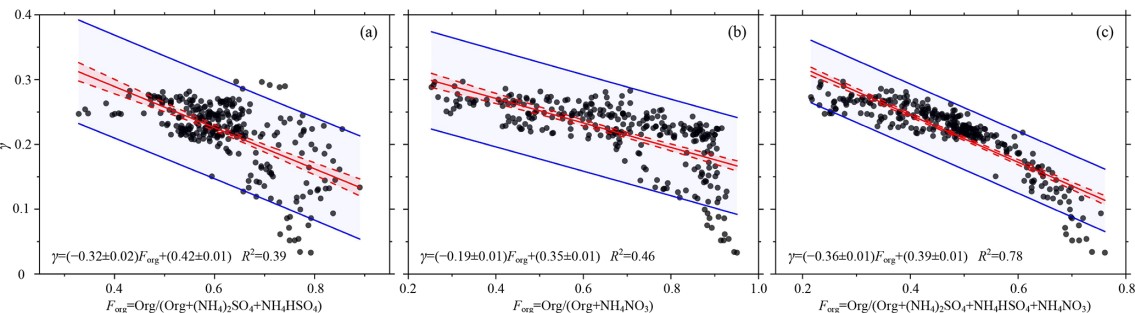

**Figure 5.** Scatter plots of $\gamma$ as a function of the relative amounts of organic and inorganic matter ($F_{org}$): **(a)** $F_{org} = Org/(Org + (NH_4)_2SO_4 + NH_4HSO_4)$, **(b)** $F_{org} = Org/(Org + NH_4NO_3)$, and **(c)** $F_{org} = Org/(Org + (NH_4)_2SO_4 + NH_4HSO_4 + NH_4NO_3)$. Solid red lines show the linear fits, dotted red lines represent the 95 % confidence levels, and solid blue lines represent the 95 % prediction bands for the fit. The linear regression function and the squared Pearson's correlation coefficient ($R^2$) are given in each panel.

sulfate in the total aerosol. This demonstrates that nitrate played a primary role in affecting aerosol hygroscopic enhancement during the study period in Beijing. To further compare the correlation between sulfate and the hygroscopicity of aerosols with that between nitrate and the hygroscopicity of aerosols, an ion-pairing scheme was used to calculate the mass concentrations of $(NH_4)_2SO_4$, $NH_4HSO_4$, and $NH_4NO_3$ in aerosols on the basis of the molar numbers of all ions (Gysel et al., 2007). The following is the ion-pairing scheme:

$$n_{NH_4NO_3} = n_{NO_3^-},$$

$$n_{H_2SO_4} = \max\left(0, n_{SO_4^{2-}} - n_{NH_4^+} + n_{NO_3^-}\right),$$

$$n_{NH_4HSO_4} = \min\left(2n_{SO_4^{2-}} - n_{NH_4^+} + n_{NO_3^-}, n_{NH_4^+} - n_{NO_3^-}\right),$$

$$n_{(NH_4)_2SO_4} = \max\left(n_{NH_4^+} - n_{NO_3^-} - n_{SO_4^{2-}}, 0\right),$$

$$n_{HNO_3} = 0, \tag{9}$$

where $n$ denotes the number of moles. Figure 5a to c show $\gamma$ as a function of $F_{org}$ (Eq. 6), where $C_i$ represents the sum of $(NH_4)_2SO_4$ and $NH_4HSO_4$, $NH_4NO_3$ and the sum of $(NH_4)_2SO_4$, $NH_4HSO_4$, and $NH_4NO_3$ mass concentrations, respectively. Overall, $\gamma$ and $F_{org}$ are negatively correlated. The coefficient of determination between $\gamma$ and $F_{org}$ ($Org/(Org + NH_4NO_3)$) (Fig. 5b) was higher than that between $\gamma$ and $F_{org}$ ($Org/(Org+(NH_4)_2SO_4 + NH_4HSO_4)$) (Fig. 5a). The coefficient of determination between $\gamma$ and $F_{org}$ ($Org/(Org + (NH_4)_2SO_4 + NH_4HSO_4 + NH_4NO_3)$) was the highest (Fig. 5c). This suggests that nitrate played a more significant role than sulfate in affecting aerosol hygroscopic growth during the study period in Beijing. In recent years, the Chinese government has made more efforts to control $SO_2$ emissions (Zhang et al., 2019), e.g., adjusting and optimizing industrial capacities. Clean fuels have also been promoted in the residential sector, with trials for using clean energy in heating in northern China carried out in all "2+26" cities and in the Fenhe River and Weihe River plains. In addition, compliance with industrial emission standards

has been strengthened. Desulfurization technology has been also applied to many heavy industrial facilities. However, China has many small-scale manufacturing enterprises, so it is much more difficult to regulate $NO_x$ emissions than $SO_2$ emissions. H. Li et al. (2109) have reported that emissions of $SO_2$ and $NO_x$ in 2017 dropped by 79.9 % and 38.1 %, respectively, from 2014 levels in Beijing, China. In 2020, $SO_2$ and primary $PM_{2.5}$ emissions dropped to 1 million tons, while $NO_x$ and volatile organic compound emissions were still 10 million tons. As a result, the decrease in $SO_2$ resulted in an increase in $NH_4NO_3$ (Morgan et al., 2010; Xu et al., 2019; Zhang et al., 2019; H. Li et al., 2019). Several previous studies focusing on megacities like Shanghai and Beijing have all suggested that the increase in nitrate mass concentration played an important role in enhancing the water content of submicron aerosols and reducing visibility under high RH conditions (Sun et al., 2012; Shi et al., 2014; Zhang et al., 2015).

Figure S10a shows the scatterplot of $\gamma$ as a function of $F_{org}(Org/(Org + (NH_4)_2SO_4 + NH_4HSO_4 + NH_4NO_3))$, where the color of the data points represents the $SO_4^{2-}/(SO_4^{2-} + SO_2)$ molar ratio. This molar ratio indicates the relative age of aerosols (Quinn et al., 2005) because by gas- and aqueous-phase oxidation processes, $SO_2$ will convert to $SO_4^{2-}$. The molar ratio of more aged aerosols is high due to the sufficient time for the conversion. The $SO_4^{2-}/(SO_4^{2-} + SO_2)$ molar ratio is low for younger aerosols. The figure shows that high $\gamma$ corresponded to high $SO_4^{2-}/(SO_4^{2-} + SO_2)$ molar ratios with a low organic matter content, while low $\gamma$ corresponded to low $SO_4^{2-}/(SO_4^{2-} + SO_2)$ molar ratios with a high organic matter content, consistent with results reported by Quinn et al. (2005) and Zhang et al. (2015). This demonstrates that the hygroscopicity of aged aerosols is higher than that of younger aerosols. Figure S10b shows that the aerosol scattering coefficients $\log_{10}(\sigma_{sp})$ were relatively low when $\gamma$ was low and the organic matter mass fraction was large. On the contrary, $\log_{10}(\sigma_{sp})$ was high when $\gamma$ was high, and the or-

ganic mass fraction was small, with a relatively large variation.

### 3.3 Deliquescence of ambient aerosols

Figure 6a shows the time series of $\eta$, with the color of the data points representing the ratio of $SO_4^{2-}$ mass concentration to $NO_3^-$ mass concentration. Figure 6b and c show the time series of wind direction and wind speed and ambient temperature and RH, respectively. Overall, deliquescence often occurred under high ambient temperature and low RH conditions (Fig. 6c). It also more easily occurred when winds with low speeds came from the south or southwest (Fig. 6b). High values of $\eta$ usually occurred when the ratio of $SO_4^{2-}$ mass concentration to $NO_3^-$ mass concentration was high (mostly higher than $\sim 4$). Figure 7a shows a relatively strong correlation between $\eta$ and the ratio of $SO_4^{2-}$ mass concentration to $NO_3^-$ mass concentration ($R = 0.62$). The blue dots represent the hysteresis index ($\eta > 0.4$) of deliquescence, showing that the corresponding ratios of $SO_4^{2-}$ mass concentration to $NO_3^-$ mass concentration were high (mostly higher than $\sim 4$). The red dots represent the hysteresis index of non-deliquescent processes, showing that the corresponding ratios of $SO_4^{2-}$ mass concentration to $NO_3^-$ mass concentration were generally less than 4. Because $PM_{2.5}$ mass concentrations were extremely low on 24 September 2019, data were noisy then, and the system error was relatively large (green dots). When these cases of large systematic errors were eliminated, the correlation between $\eta$ and the ratio of $SO_4^{2-}$ to $NO_3^-$ mass concentration increased ($R = 0.69$). In the North China Plain, Kuang et al. (2016) also observed a similar deliquescent phenomenon associated with $(NH_4)_2SO_4$, while in Ny-Ålesund, Norway, this deliquescent phenomenon was related to sea salt (Zieger et al., 2010).

According to observational results, there are two environmental conditions of consequence for ambient aerosol deliquescence:

1. high ambient temperature and low ambient RH

2. relatively good air quality and solar illumination.

Concerning the first condition, Cheung et al. (2015) proposed an indicator describing the neutralization extent of aerosols, i.e., the molar ratio, $MR = (NH_4^+ - NO_3^-) / SO_4^{2-}$. The MR value of the non-deliquescent process was always the lowest in their study. Their results indicate that when the nitrate content of ambient aerosols was slightly high, and the sulfate content was low, aerosol particles did not exhibit deliquescence resulting from acidic sulfate. Here, only when the mass concentration ratio of sulfate to nitrate was high (mostly higher than $\sim 4$) did ambient aerosol particles exhibit deliquescence. Figure 8a and b illustrate the diurnal variations in nitrate, along with ammonium and sulfate. Both the mass concentration and mass fraction of nitrate reached their largest and smallest values in the early morning and in the

afternoon, respectively. It was closely correlated with the diurnal variation in ambient temperature and RH, where RH gradually rose as the temperature gradually decreased from the late afternoon to the early morning of the next day, followed by a reversal of the trend into the late afternoon of the next day (Fig. 6c). Morino et al. (2006) and Wang et al. (2009) found that the formation of nitrate requires low temperatures and high RH, conditions favorable for the conversion of gaseous nitric acid to solid-phase nitrate. Ambient aerosol deliquescence is thus closely related to the environmental conditions of high temperature and low humidity. Note that secondary sulfates are usually produced by photochemical reactions so that the sulfate content of aerosols gradually accumulates from late morning to late afternoon (Huang et al., 2010; Sun et al., 2012). Generally speaking, the ambient temperature was higher, and the humidity was lower in the afternoon, so high values of $SO_4^{2-} / NO_3^-$ mostly occurred in the afternoon. This may also explain why the deliquescent cycle occurred most frequently in the afternoon, as shown in Fig. 7b.

The blue-shaded zones in Fig. 1a and f show that deliquescence usually occurred when the air quality was good. To better explain this phenomenon, the observation period was divided into a very clean period ($PM_{2.5} \le 35 \,\mu g \, m^{-3}$) and a moderately polluted period ($PM_{2.5} > 75 \,\mu g \, m^{-3}$). The difference between $f(RH = 85\%, 525 \, nm)$ and $f(RH = 80\%, 525 \, nm)$, i.e., DF, or $f(RH = 85\%, 525 \, nm)$ minus $f(RH = 80\%, 525 \, nm)$, was calculated. Figure 8c shows the values of DF during a very clean period and a moderately polluted period. The DF of deliquescence is larger than that of non-deliquescence because the $f(RH)$ of deliquescence jumps suddenly as the RH increases from 80% to 85%, while the $f(RH)$ of non-deliquescence increases smoothly as the RH increases. The values of the particle light scattering enhancement factor at high RH for deliquescent processes were usually higher than those for non-deliquescent processes, consistent with the strong hygroscopicity of $(NH_4)_2SO_4$. Figure 8c shows that the diurnal range of DF during the clean period was larger than that during the moderately polluted period. In particular, from late morning to late afternoon, the DF during the clean period was much higher than that during the moderately polluted period. Figure 8d shows the diurnal variation in $SO_4^{2-} / NO_3^-$. The ratio $SO_4^{2-} / NO_3^-$ during the clean period was higher than that during the moderately polluted period, especially from late morning to late afternoon. Overall, good air quality and solar illumination were conducive to photochemical reactions so that more secondary sulfate could be generated during the day, facilitating the deliquescence of ambient aerosols.

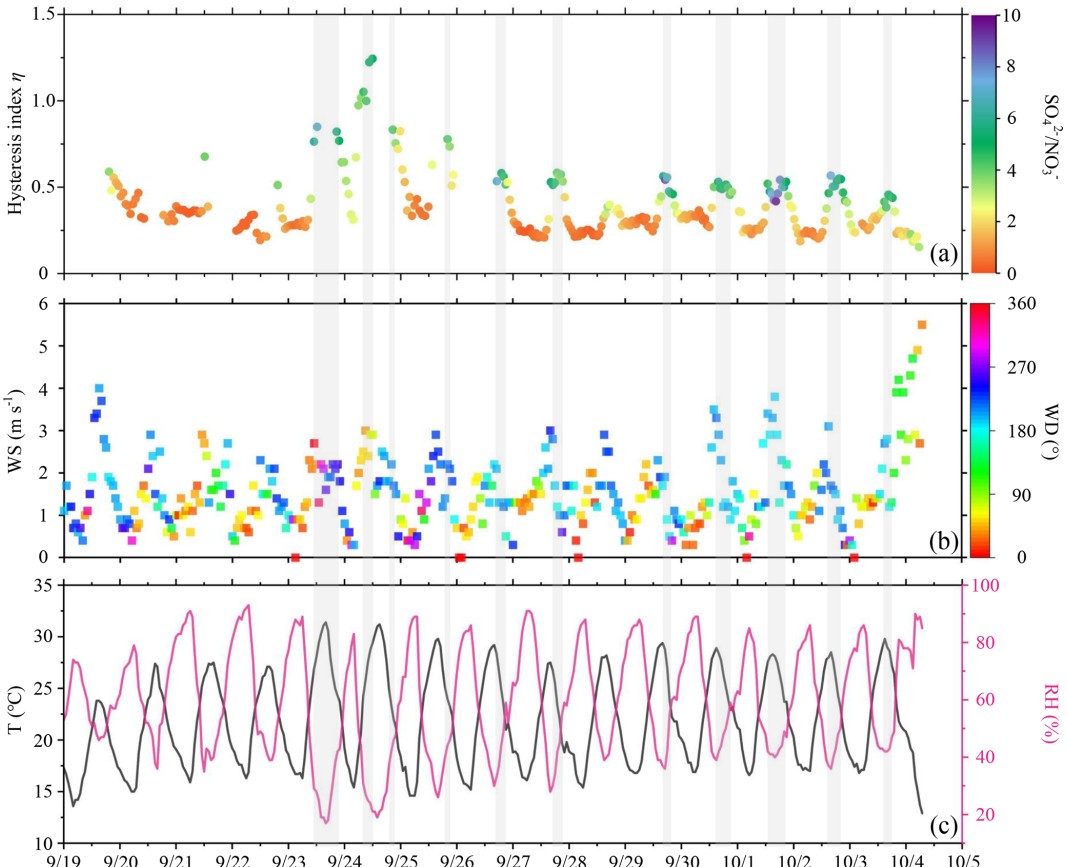

**Figure 6.** Time series of **(a)** hysteresis index $\eta$ colored by the $SO_4^{2-}/NO_3^-$ mass concentration ratio, **(b)** wind speed (WS; unit: $m\,s^{-1}$) colored by wind direction (WD; unit: °), and **(c)** ambient temperature ($T$; unit: °C) and relative humidity (RH; unit: %). The segments of the time series with a grey background represent the processes of deliquescence. The date in this figure is in the month/day format.

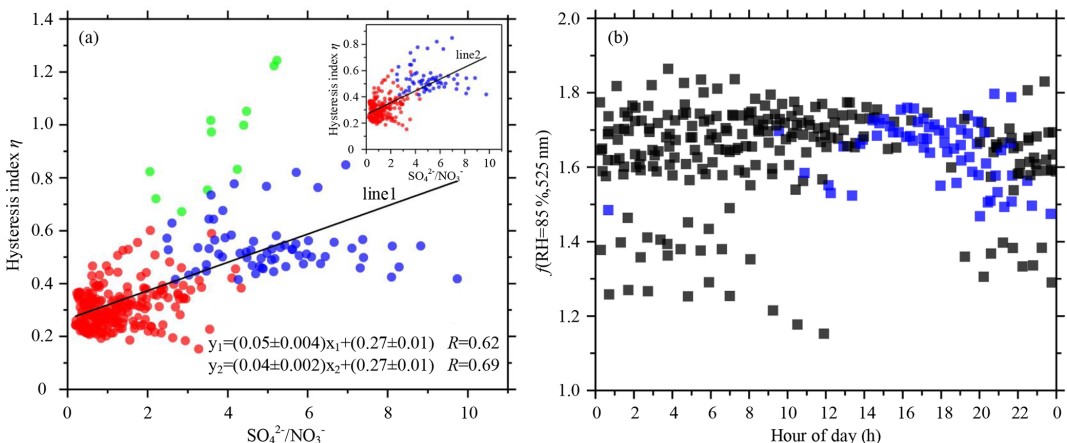

**Figure 7. (a)** Scatter plot of hysteresis index $\eta$ as a function of the $SO_4^{2-}/NO_3^-$ mass concentration ratio. Blue dots represent deliquescence and red dots non-deliquescent processes. Green dots represent those data points with high systematic errors. The inset figure shows the scatter plot excluding the green dots. **(b)** Scatter plots of the observed $f\,(RH = 85\,\%, 525\,nm)$ values for non-deliquescent (black) and deliquescent (blue) cycles. The linear regression functions and the Pearson's correlation coefficients ($R$) are given in the bottom right corner of **(a)** for each fitted line.

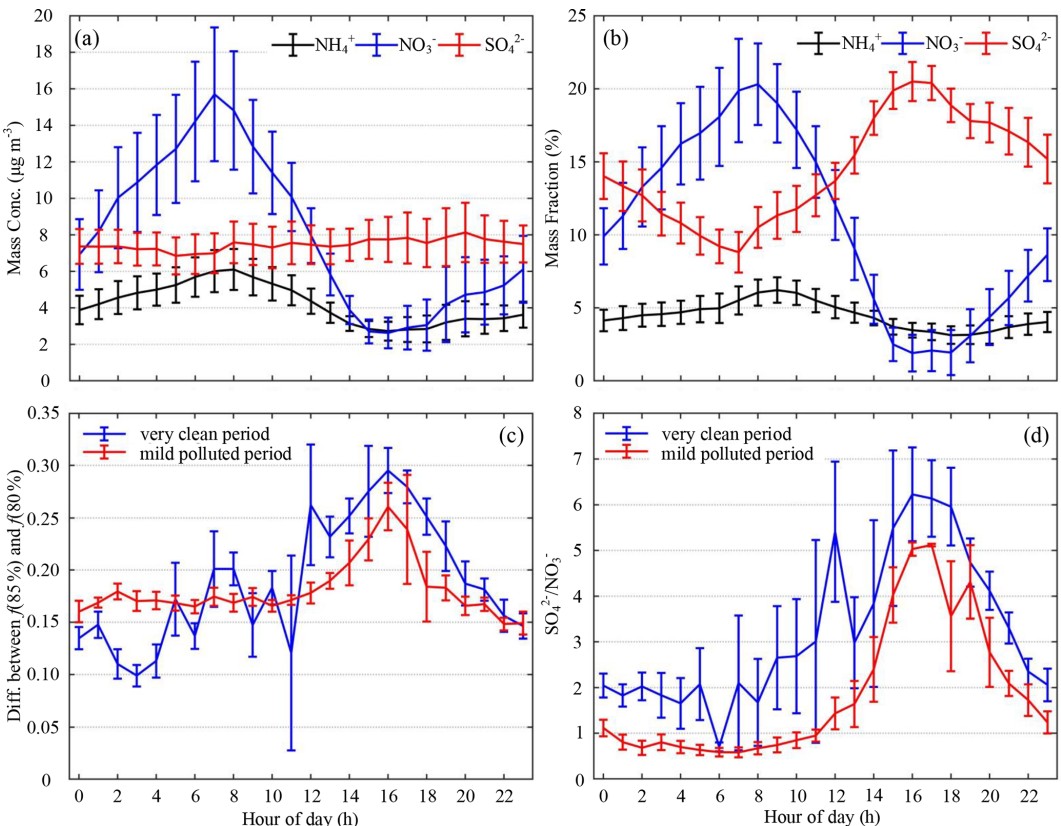

**Figure 8.** Average diurnal cycles of **(a)** mass concentration (unit: $\mu g\,m^{-3}$) and **(b)** mass fraction (unit: %) of ammonium (black curves), nitrate (blue curves), and sulfate (red curves) measured by the ACSM. **(c)** The difference between $f(RH = 85\,\%, 525\,nm)$ and $f(RH = 80\,\%, 525\,nm)$ and **(d)** the mass concentration ratio of sulfate to nitrate ($SO_4^{2-}/NO_3^-$) during the clean period (blue curve) and the moderately polluted period (red curve). Vertical lines are the standard deviations.

## 3.4 Parameterizations of $f(RH)$

### 3.4.1 Parameterization with the equation $f(RH) = 1 + m \times RH^n$

Many empirical expressions have been presented to parameterize $f(RH)$ (Kotchenruther and Hobbs, 1998; Carrico et al., 2003; Pan et al., 2009; Fierz-Schmidhauser et al., 2010a; Chen et al., 2014; Brock et al., 2016; Titos et al., 2016; Kuang et al., 2017). The following is the two-parameter scheme introduced by Kotchenruther and Hobbs (1998):

$$f(RH) = 1 + m \times RH^n. \tag{10}$$

The parameter $m$ determines the largest value of $f(RH = 100\,\%)$, and the parameter $n$ dominates the magnitude of the scattering enhancement, reflecting the curvature of the humidogram.

Deliquescence was frequently observed during the entire measurement campaign. In total, 294 cycles of $f(RH)$ were measured, and 47 cycles (16 % of all cycles) showed clear deliquescence (Fig. 9d, e). All $f(RH)$ curves were thus first classified into deliquescent curves and non-deliquescent curves. After averaging PM$_{2.5}$ concentrations of the corresponding cycles, all non-deliquescent curves were further divided into clean (PM$_{2.5} \leq 35\,\mu g\,m^{-3}$), moderately polluted ($35\,\mu g\,m^{-3} < $ PM$_{2.5} \leq 75\,\mu g\,m^{-3}$), and polluted (PM$_{2.5} > 75\,\mu g\,m^{-3}$) categories. The deliquescent curves were divided into clean and moderately clean categories only because deliquescence mainly occurred under good air quality conditions. For cycles without deliquescence (Fig. 9a–c), the measured values were fitted using Eq. (10). For cycles with deliquescence (Fig. 9d, e), $f(RH)$ increased smoothly under low RH conditions then increased sharply. Under low and high RH conditions, the fitted $f(RH)$ values were usually lower than observed values, but the slopes of the two curves were similar. However, when RH approached $\sim 80\,\%$ where $f(RH)$ sharply increased, the fitted $f(RH)$ values were usually higher than observed values, with different slopes of the two curves. Therefore, segment fitting (Eq. 12) was applied in the parameterization of deliquescent $f(RH)$. The deliquescence observed in our study was primarily caused by $(NH_4)_2SO_4$ in ambient aerosols. The DRH of the pure $(NH_4)_2SO_4$ aerosols generated in the laboratory was 80.07 %, measured by our high-resolution hu-

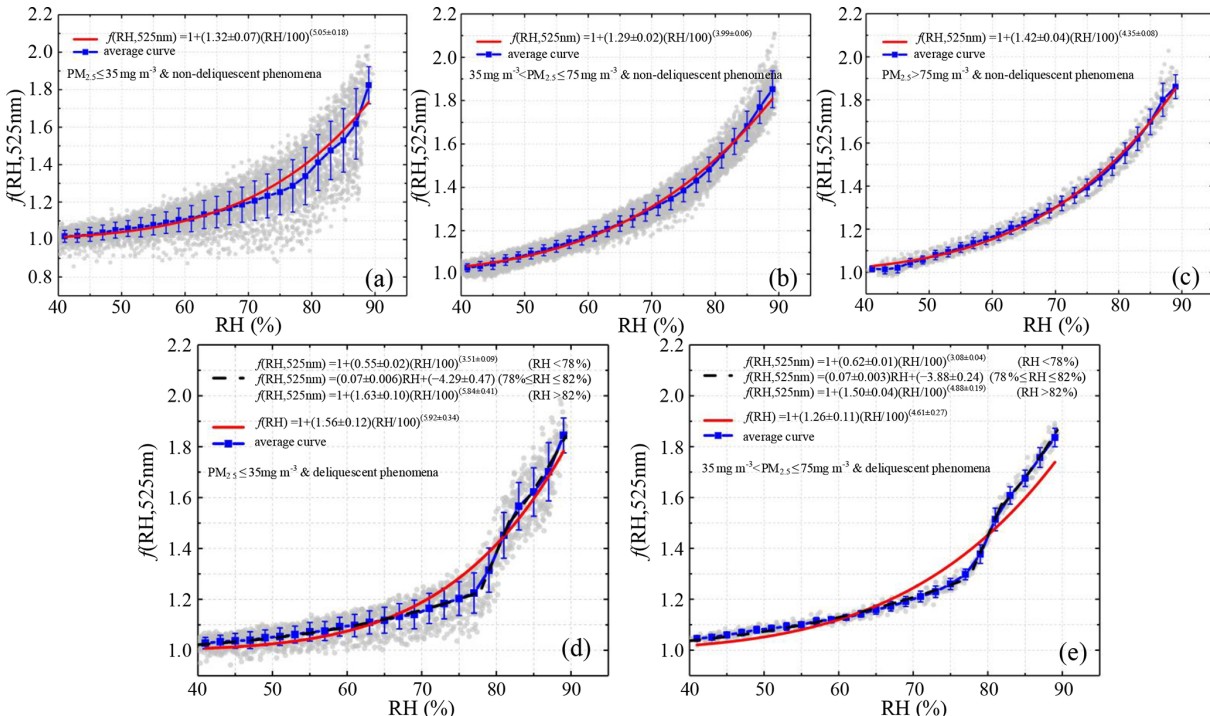

**Figure 9.** Fitted $f(RH)$ of non-deliquescent curves **(a–c)** and deliquescent curves **(d–e)** for different pollution levels. Solid red lines represent fitted curves parameterized by Eq. (10). Dotted black lines represent the curves fit by segment function Eq. (12), and the solid blue lines represent the average curves.

midified nephelometer system (Fig. S4). Because the DRH of all deliquescence in this study, according to statistics, was mainly distributed between 78 % and 80 %, Eq. (10) was applied to fit $f(RH)$ values when RH ≥ 82 % or RH ≤ 78 %. For 78 % < RH < 82 %, $f(RH)$ values were parameterized by Eq. (11):

$$f(RH) = a \times RH + b. \tag{11}$$

Parameter $a$ determines the degree of $f(RH)$ jumps during deliquescence, i.e., the slope of the linear fitting line. Parameter $b$ determines the $f(RH, 525\,nm)$ value before deliquescence (RH = 78 %) and after deliquescence (RH = 82 %).

$$\begin{cases} f(RH) = 1 + m \times RH^n & RH \leq 78\% \\ f(RH) = a \times RH + b & 78\% < RH < 82\% \\ f(RH) = 1 + m \times RH^n & RH \geq 82\% \end{cases} \tag{12}$$

Table 2 summarizes the parameters $m$ and $n$ used in this study and in similar studies. For the non-deliquescence case, the parameter $n$ for the clean period was the largest (~ 27 % larger than that for the moderately polluted period and ~ 16 % larger than that for the polluted period). The difference between the clean and moderately polluted parameter $m$ was small, while the parameter $m$ in the polluted period was about 0.1 larger than that in the other two periods. As a result, $f(RH = 85\%, 525\,nm)$ was the largest in the polluted period and the smallest in the clean period. For the deliquescence case, the fitted parameters $m$ and $n$ for the whole

RH range (40 % < RH < 90 %) were much larger than the $m$ and $n$ for RH < 78 % and slightly larger than the $m$ and $n$ for RH > 82 %. If Eq. (10) was used to parameterize the whole RH range of the deliquescence curves instead of segment fitting, bigger differences between the parameterized fitting results and measured values would occur. This would cause greater uncertainties in the model simulation of aerosol hygroscopicity. Also, there was no significant difference between parameter $a$ under clean ($a = 0.07 \pm 0.006$) and moderately polluted ($a = 0.07 \pm 0.003$) conditions, indicating that the trend and amplitude of the jump growth for aerosol deliquescence are consistent under these two environmental conditions. However, the absolute value of parameter $b$ under clean conditions ($b = -4.29 \pm 0.47$) was higher than that under moderately polluted conditions ($b = -3.88 \pm 0.24$), indicating that the DRH for clean periods was slightly higher than that for moderately polluted periods.

### 3.4.2 Steepness of humidograms

The steepness index proposed by Zhang et al. (2015), $\tau$, is defined as

$$\tau = \frac{f'(80\%)}{f'(60\%)} - 1 = \left(\frac{4}{3}\right)^{n-1} - 1 \tag{13}$$

to quantitatively describe the change in the curvature of the humidogram curves; $f'(80\%)$ and $f'(60\%)$ are the deriva-

**Table 2.** Curve-fitting parameters of $f(\mathrm{RH})$ at 525 nm for different aerosol types using Eqs. (10) and (12).

| Classification | | | $m$ | $n$ | $a$ | $b$ | Reference |
|---|---|---|---|---|---|---|---|
| Non-deliquescence | Very clean | | $1.32 \pm 0.07$ | $5.05 \pm 0.18$ | | | This study |
| | Moderately polluted | | $1.29 \pm 0.02$ | $3.99 \pm 0.06$ | | | |
| | Polluted | | $1.42 \pm 0.04$ | $4.35 \pm 0.08$ | | | |
| Deliquescence | Very clean | RH < 78 % | $0.55 \pm 0.02$ | $3.51 \pm 0.09$ | | | |
| | | RH > 82 % 78 % < RH < 82 % | $1.63 \pm 0.10$ | $5.84 \pm 0.41$ | $0.07 \pm 0.006$ | $-4.29 \pm 0.47$ | |
| | | 40 % < RH < 90 % | $1.92 \pm 0.41$ | $6.96 \pm 1.63$ | | | |
| | Moderately polluted | RH < 78 % | $0.62 \pm 0.01$ | $3.08 \pm 0.04$ | | | |
| | | RH > 82 % 78 % < RH < 82 % | $1.50 \pm 0.04$ | $4.88 \pm 0.19$ | $0.07 \pm 0.003$ | $-3.88 \pm 0.24$ | |
| | | 40 % < RH < 90 % | $1.63$ | $5.61$ | | | |
| Clean | | | $1.20 \pm 0.06$ | $6.70 \pm 0.27$ | | | Pan et al. (2009) |
| Polluted | | | $2.30 \pm 0.03$ | $6.27 \pm 0.10$ | | | |
| Dust | | | $0.64 \pm 0.04$ | $5.17 \pm 0.4$ | | | |
| Locally polluted | | | $1.24 \pm 0.29$ | $5.46 \pm 1.90$ | | | Zhang et al. (2015) |
| Northerly polluted | | | $1.20 \pm 0.21$ | $3.90 \pm 1.27$ | | | |
| Dust-influenced | | | $1.02 \pm 0.19$ | $4.51 \pm 0.80$ | | | |

tives of the fitted curve of $f(\mathrm{RH})$ given in Eq. (10) at the two different RH values.

A low $\tau$ means that the curvature of the humidogram is small, and a high $\tau$ means that the slopes of the curve from low to high RH sharply change. Figure S11 shows the scatter plot of $\tau$ as a function of the nitrate mass fraction, colored by the sulfate mass fraction. In the 0 %–15 % nitrate mass fraction range, $\tau$ decreased sharply as the nitrate mass fraction increased, demonstrating that the curvature of the humidogram became smaller. For nitrate mass fractions larger than 15 %, $\tau$ stabilized to a constant value of $\sim 1.6$, and the curvature of the humidogram was much smaller.

## 4 Conclusions

Direct measurements of aerosol hygroscopicity, as expressed by $f(\mathrm{RH},\lambda)$, were carried out at a site in the southern urban edge of Beijing, aimed at investigating the effect of aerosol water uptake on particle light-scattering properties. The mass concentrations of aerosol chemical components were measured simultaneously by the ACSM and AE-33. Also measured were other aerosol parameters, such as the light absorption coefficient and the mass concentration of $\mathrm{PM}_{2.5}$. In total, 294 cycles of $f(\mathrm{RH})$ were measured, and 47 cycles (16 % of all cycles) showed clear deliquescence.

The proportion of components making up the chemical composition of aerosols is key to influencing $f(\mathrm{RH})$. In general, $f(\mathrm{RH})$ had a strong positive correlation with the proportion of inorganic matter and a negative correlation with the proportion of organic matter and eBC. High values of $f(\mathrm{RH})$ usually occurred for aged aerosols whose mass fraction of organic matter was small. Low values of $f(\mathrm{RH} = 85\,\%, 525\,\mathrm{nm})$ often occurred for primary aerosols with more organic matter. Furthermore, when the mass frac-

tion of inorganic matter was smaller than 40 %, the positive correlation between $f(\mathrm{RH} = 85\,\%, 525\,\mathrm{nm})$ and inorganic matter was much stronger. A similar phenomenon was also found for the negative correlation between organic matter and $f(\mathrm{RH} = 85\,\%, 525\,\mathrm{nm})$. The compensating effect between the size and chemical composition of aerosol may be one of the main reasons for this phenomenon. High mass concentrations of ambient aerosols and insufficient water vapor in the sample flow may be the other two reasons. Nitrate also played a more significant role in affecting aerosol hygroscopic growth than sulfate in Beijing.

Favorable meteorological conditions for aerosol deliquescence were high ambient temperature and low RH, as well as relatively good air quality and strong solar illumination. High temperatures and low RH levels were not conducive to the formation of nitrate, while good air quality and light conditions were conducive to photochemical reactions so that more secondary sulfates could be generated. Only when the ratio of the sulfate mass fraction to the nitrate mass fraction was greater than $\sim 4$ did the deliquescence phenomenon of ambient aerosols easily occur.

All humidograms were first classified as either deliquescent or non-deliquescent. The two kinds of humidograms were further classified according to the mass concentration of $\mathrm{PM}_{2.5}$. The two-parameter scheme, $f(\mathrm{RH}) = 1 + m \times \mathrm{RH}^n$, introduced by Kotchenruther and Hobbs (1998), was applied to fit the non-deliquescent $f(\mathrm{RH})$. The deliquescent $f(\mathrm{RH})$ was parameterized by segment functions (Eq. 12). For the deliquescence case, the fitted parameters $m$ and $n$ for the whole RH range (40 % < RH < 90 %) were much larger than the parameters for RH < 78 % and higher than the parameters for RH > 82 %. This demonstrates that large errors would be incurred if only Eq. (10) was used to parameterize the whole RH range of deliquescence curves. The piecewise parameterization scheme (Eq. 12) is a better fit for humido-

grams representing deliquescence to reduce uncertainties in the model simulation of aerosol hygroscopicity. The curvature of the $f(RH)$ humidogram, described by the steepness index, decreased sharply as the nitrate mass fraction increased within the range of 0 %–15 %. When the nitrate mass fraction was larger than 15 %, the steepness index remained constant ($\sim 1.6$).

*Data availability.* The data used in this paper can be downloaded online (https://doi.org/10.11922/sciencedb.00785) TS6.

*Supplement.* The supplement related to this article is available online at: https://doi.org/10.5194/acp-21-1-2021-supplement. TS7

*Author contributions.* ZL and PY designed the field experiment. ZL, PY, and RR determined the main goal of this study. RR processed the measurement data and prepared this paper with contributions from all co-authors. PY, HW, and YW provided technical guidance for instrumentation. MC copy-edited the article. WW, XJ, YL, and DZ CE6 participated in the implementation of this experiment.

*Competing interests.* The authors declare that they have no conflict of interest.

*Acknowledgements.* This work was supported by the National Key R&D Program of China (no. 2017YFC1501702), the National Science Foundation of China (nos. 91544217, 42005067), the Open Fund of State Key Laboratory of Remote Sensing Science (no. 202015), and the Guangdong Basic and Applied Basic Research Fund Committee (2020B1515130003).

*Financial support.* This research has been supported by the NAME OF FUNDER (grant no. GRANT AGREEMENT NO). TS8

*Review statement.* This paper was edited by Paul Zieger and reviewed by two anonymous referees.

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

TS2    Please confirm.

TS3    Please add last access date.

TS4    Please add last access date.

TS5    Please confirm.

TS6    Please provide a reference list entry including creators, title, and date of last access.

TS7    Please send a new supplement as a *.pdf without the title, authors, correspondence author, etc. as we will generate a supplement title page during publication (with a citation including the DOI), which will contain this information.

TS8    Please note that there is funding information given in the acknowledgements but you have not indicated any funding upon manuscript registration. Therefore, we were not able to complete the financial support statement. Please provide the missing information and double-check your acknowledgements to see whether repeated information can be removed from the acknowledgement. Thanks.

TS9    Please add total pages.

TS10    Please add total pages.