# Peer review of "Measurement report: The effect of aerosol chemical composition on light scattering due to the hygroscopic swelling effect"

_Atmospheric Chemistry and Physics, 2020_

## Author Comment (AC1)

This manuscript presents the results of an intensive field campaign of two week duration in an area south of Beijing. The manuscript focuses on the study of the effect of aerosol hygroscopicity in the aerosol light scattering coefficient. Despite the short measurement period, the data collected are interesting and the results are of scientific significance. Nevertheless, I have some minor comments that need to be clarify by the authors.

Line 78: Briefly describe the humidifying scheme by Carrico et al. (1998)
**Reply:** The humidifying scheme by Carrico et al. (1998) is described in the manuscript as: "The water vapor controlled by the temperature of the liquid water in the outer annulus of the tube passes through a Teflon membrane, humidifying aerosols in the inner tube (Carrico et al., 1998). The temperature of the liquid water was controlled by adjusting the power of the water baths."
[Lines 84-86]

Line 80: Was the tandem nephelometer calibrated with ammonium sulphate or other salt of known hygroscopicity? This is highly recommended to assure that the system is functioning correctly and that the RH inside the nephelometer chamber is correct (see Burgos et al. (2019) and Fierz-Schmidhauser et al. 2010b).
**Reply:** The tandem nephelometer in this study was calibrated with ammonium sulphate, whose deliquescence relative humidity (DRH) was 79.9±0.5% at 298 K. The DRH of pure ammonium sulfate aerosols generated in the laboratory was 80.367%, measured by our high-resolution humidified nephelometer system (Fig. R1). This shows that the RH inside the nephelometer chamber was correct and that the system was functioning properly.

[Figure]

**Figure R1: Deliquescence results of the pure ammonium sulfate aerosol generated in the laboratory.**
[Lines 108-111]

Line 80: Were the nephelometers operated with or without the kalman filter option?
**Reply:** Since the RH of aerosols inside the nephelometers was constantly changing, and real measured data at every moment was needed, the nephelometers operated without the Kalman filters.
[Lines 91-92]

Line 90: The equation to calculate the dew point temperature is not the most common one. I looked in the references provided by the authors but Kuang et al. (2017) doesn't state which formula they use to calculate T dew point and the reference of Liu and Zhao (2016) is in Chinese. Please, use appropriate references for this formula.
Also, it would be interesting to see the comparison between RHin, RHoutlet and RHcalculated.

**Reply:** We have revised the references in the manuscript. Equations (1) and (2) are from a Ph.D. thesis in Chinese (L. Zhang, 2017), explained as follows. Because the RH levels measured by the probe built into the optical chamber of the wet nephelometer ($RH_{chamber}$) was imprecise, two calibrated RH and temperature probes were placed at the inlet and outlet of the wet nephelometer, obtaining 1-min averages of RH and temperature (Fig. R2). We used Vaisala HMP110 probes with accuracies of $\pm 0.2$℃ for the 0–40℃ temperature range and $\pm 1.5\%$ RH and $\pm 2.5\%$ RH for the 0–90% and 90–100% RH ranges, respectively. As shown in Figure R2 (b), the temperatures measured by these probes were different. However, in principle, the dew point temperatures ($T_{dew-point}$) at these positions are all the same. Since the RH and temperature probes at the outlet of the wet nephelometer ($RH_{outlet}$ and $T_{outlet}$) were less affected by the humidifier, $RH_{outlet}$ and $T_{outlet}$ were used to calculate $T_{dew-point}$ at this position using Eq. (1):

$$T_{dew-point} = RH_{outlet}^{\frac{1}{8}}(112 + 0.9T_{outlet}) + 0.1T_{outlet} - 112. \qquad (1)$$

We assume that $T_{dew-point}$ was approximately the same as that in the optical chamber of the wet nephelometer. Based on the temperature in the optical chamber ($T_{chamber}$) and $T_{dew-point}$, the actual RH in the optical chamber ($RH_{chamber}$) can be calculated by rearranging Eq. (1), i.e.,

$$RH_{chamber} = \left(\frac{112 - 0.1T_{chamber} + T_{dew-point}}{112 + 0.9T_{chamber}}\right). \qquad (2)$$

[Lines 94-107]

Figures R2 (a) and (b) show time series of $RH_{inlet}$, $RH_{outlet}$, and $RH_{chamber}$ and $T_{inlet}$, $T_{outlet}$, and $T_{chamber}$, respectively. Temperatures measured by the three probes are significantly different. Since the RH and temperature probes at the inlet of the wet nephelometer is very close to the humidifying tube, $T_{inlet}$ is influenced by the humidifier. Also, $T_{outlet}$ is lower than $T_{chamber}$. Figure R1 (a) shows that $RH_{chamber}$ is lower than $RH_{outlet}$.

Figure R2: Time series of (a) $RH_{in}$, $RH_{outlet}$, and $RH_{chamber}$ and (b) $T_{in}$, $T_{outlet}$, and $T_{chamber}$ on 29 November 2019. Times in this figure are in the hour/minute format.

[Figure]

Figure R3: Diagram of the structure of the wet nephelometer. The three red triangles show the locations of the three sets of RH and temperature probes.

Line 105: I really don't understand how the f(RH) is calculated. Why f(RH) is normalized? What is the reason behind this? Also, f(RH>40%) is averaged over what? Whole dataset, each scan?
Then, in line 108 it is said that f(RH>40%) is 1. This is true for all the observations? Is it exactly 1? This calculation needs clarification.
**Reply:** In this study, we assumed that the aerosol is in the dry state when RH < 40%. So $f$(RH) should theoretically equal 1 when RH is lower than 40%. This was what was meant by the sentence: "Here, f(RH < 40%) equals 1." However, due to systematic errors and the differences in RH measured synchronously by the dry nephelometer and the wet nephelometer, respectively, measured $f$(RH < 40%) have small fluctuations and are not equal to 1. This is why $f$(RH) was normalized. Not $f$(RH > 40%), but the corrected coefficient ($f$ (RH < 40%)$_{averaged}$), which was averaged over the whole dataset of RH < 40%. $f$(RH > 40%) was then normalized using Eq. (4).
[Lines 124-129]

Line 112: The absorption coefficient is measured at 7 wavelengths, the absorption coefficient at 520 nm is more appropriate than using the absorption coefficient at 880 nm and then convert it to 525 nm.
**Reply:** Done. The single-scattering albedo ($\omega_{0(525nm)}$) has also been recalculated. Also updated were Figs. 2b, 2c, 3a and Table 1.
[Lines 130-132]

Eq 6: So, only $f$(RH=85%) is used to calculate gamma? If the frh measurements are performed at scanning RH it can be retrieved from a potential fit using the whole RH range, which will have less errors than using a single RH point (see Zieger et al. 2010, Titos et al., 2016).
**Reply:** We appreciate the suggestion. Previously, only $f$(RH = 85%) and $f$(RH = 40%) were used to calculate $\gamma$, which has now been revised. The parameter $\gamma$ is now retrieved from the following $f$(RH) parametrization scheme: $f$(RH) = $(1 - RH)^{-\gamma}$ (i.e., Eq. (8)), using the whole RH range (generally from ~ 40% to ~ 90%). Figures 7 and S7 have been updated accordingly.

Line 125: Include a reference to Zieger et al. 2010, who firstly introduced the hysteresis index.
**Reply:** The reference is added.

Eq9: Actually, what it is here called g, it is usually referred as gamma.
**Reply:** Thank you.

Eq. 8: The RH range used to identify deliquesce is very narrow and can miss deliquescence processes occurring at slightly different RH. Maybe consider the procedure of Zieger et al.
**Reply:** We appreciate the suggestion. The deliquescence observed in this study mainly resulted from the ammonium sulfate in ambient aerosols. The deliquescence RH (DRH) of pure ammonium sulfate aerosols generated in the laboratory was 80.367%, measured by our high-resolution humidified nephelometer system (Fig. S8). Forty-seven cycles of $f$(RH) (16% of all cycles) in this study show clear deliquescence, and the DRH of all deliquescence are mainly distributed between 78% and 80%. So the RH range used to identify deliquescence in this paper (78% < RH < 82%) does not miss deliquescent processes. In addition, if we use a larger RH to identify deliquescence, for example, 75% < RH < 85%, less data would be available to calculate $\gamma_{>85\%}$ on each hydration branch.

Fig7: Why not consider all measured species, including NH4+, Cl- and BC? Is the organic mass fraction defined differently than in Figure 6?
**Reply:** Since chloride ions ($Cl^-$) accounted for less than 1% of submicron aerosols during the entire measurement period, the influence of $Cl^-$ on the hygroscopic enhancement factor of aerosols is essentially negligible. In addition, chemical species consisting of $Cl^-$ are hardly determined. Therefore, the influence of $Cl^-$ on $\gamma$ has not been considered in Fig. 7. As for BC, its influence on $f$(RH = 85%, 525 nm) has been analyzed in Fig. 5d. Since $\gamma$ is the parameter that can replace $f$(RH) over a wider RH range, it is not necessary to discuss the correlation between BC and $\gamma$ again in Fig. 7. Considering the importance of the ammonia ion ($NH_4^+$) on the $f$(RH) of aerosols in previous studies (Zieger et al., 2010; L. Zhang et al., 2015), it is a helpful suggestion to take $NH_4^+$ into account. So an ion-pairing scheme was conducted to calculate the mass concentration of ammonium sulfate ($(NH_4)_2SO_4$), ammonium bisulfate ($NH_4HSO_4$), and ammonium nitrate ($NH_4NO_3$) in aerosols on the basis of the molar numbers of all ions (Gysel et al., 2007). The following is the ion-pairing scheme:

$$n_{NH_4NO_3} = n_{NO_3^-}$$
$$n_{H_2SO_4} = \max\left(0, n_{SO_4^{2-}} - n_{NH_4^+} + n_{NO_3^-}\right)$$
$$n_{NH_4HSO_4} = \min\left(2n_{SO_4^{2-}} - n_{NH_4^+} + n_{NO_3^-}, n_{NH_4^+} - n_{NO_3^-}\right) \qquad (9)$$
$$n_{(NH_4)_2SO_4} = \max\left(n_{NH_4^+} - n_{NO_3^-} - n_{SO_4^{2-}}, 0\right)$$
$$n_{HNO_3} = 0,$$

where $n$ denotes the number of moles. Figures 7a-c now show $\gamma$ as a function of $F_{org}$ (Eq. 6), where $C_i$ represents the sum of $(NH_4)_2SO_4$ and $NH_4HSO_4$, $NH_4NO_3$, and the sum of $(NH_4)_2SO_4$, $NH_4HSO_4$, and $NH_4NO_3$ mass concentrations, respectively. Figure 7 has been revised accordingly.
The organic mass fraction defined in Fig. 7 is the same as that in Fig. 6.

Fig8: Use same color for WD from north (360º and 0º)
**Reply:** We have revised Fig. 8b accordingly.

Line 220: Don't understand the reasoning, which marine aerosols do the authors refer to?
**Reply:** The absolute values of both slopes and corresponding standard deviations (0.80±0.04 and 1.00±0.06 for $f$(RH = 85%, 525 nm) as a function of inorganic and organic matter mass fractions, respectively) shown in Fig. 6 were similar to those reported in Lin'an, China (0.96±0.02 and 1.20±0.04, respectively; L. Zhang et al., 2015) but much lower than those observed at Melpitz, Germany (2.2±0.078 and 3.1±0.1, respectively; Zieger et al.,

2014). This might be because the $f$(RH = 85%, 525 nm) measured in Melpitz, Germany, was much higher than that in Lin'an and Beijing. Ambient aerosols in Melpitz, Germany, were affected by sea salt, like sodium chloride, transported from the North Sea and highly hygroscopic. Marine aerosols have a higher hygroscopicity than aerosols influenced more by human activity.
[Lines 234-240]

Line 245: Do the authors refer to an instrument artefact due to water depletion?
**Reply:** Yes, we do, because the amount of water vapor passing through the Teflon membrane to humidify aerosols in the inner tube was finite at a specific temperature. The amount of water vapor may be insufficient when the proportion of inorganic matter is very high.

Line 260: It is not that in the previous studies the role of NO3- was not as important as in the present study. Quinn et al. (2005) didn't look at NO3-, their organic mass fraction was calculated using only SO42- as inorganic component. Why do the authors don't include NH4+? Previously they stated the importance of ammonia, but here it is not included. See the relationships obtained by Zieger et al. and Zhang et al.
**Reply:** The unclear statements have been revised. As we stated in the response to the suggestion given by the reviewer for Fig. 7, an ion-pairing scheme was conducted to calculate the mass concentrations of ammonium sulfate ($(NH_4)_2SO_4$), ammonium bisulfate ($NH_4HSO_4$), and ammonium nitrate ($NH_4NO_3$) in aerosols on the basis of the molar numbers of all ions (Gysel et al., 2007). The following is the ion-pairing scheme:

$$n_{NH_4NO_3} = n_{NO_3^-}$$
$$n_{H_2SO_4} = \max(0, n_{SO_4^{2-}} - n_{NH_4^+} + n_{NO_3^-})$$
$$n_{NH_4HSO_4} = \min(2n_{SO_4^{2-}} - n_{NH_4^+} + n_{NO_3^-}, n_{NH_4^+} - n_{NO_3^-}) \qquad (9)$$
$$n_{(NH_4)_2SO_4} = \max(n_{NH_4^+} - n_{NO_3^-} - n_{SO_4^{2-}}, 0)$$
$$n_{HNO_3} = 0,$$

where $n$ denotes the number of moles. The updated Figs. 7a-c show $\gamma$ as a function of $F_{org}$ (Eq. 6), where $C_i$ represents the sum of $(NH_4)_2SO_4$ and $NH_4HSO_4$, $NH_4NO_3$, and the sum of $(NH_4)_2SO_4$, $NH_4HSO_4$, and $NH_4NO_3$ mass concentrations, respectively. Overall, $\gamma$ and $F_{org}$ are negatively correlated. The coefficient of determination between $\gamma$ and $F_{org}$ (Org/(Org+$NH_4NO_3$)) (Fig. 7b) was higher than that between $\gamma$ and $F_{org}$ (Org/(Org+ $(NH_4)_2SO_4$ + $NH_4HSO_4$)) (Fig. 7a). The coefficient of determination between $\gamma$ and $F_{org}$ (Org/(Org+$(NH_4)_2SO_4$ + $NH_4HSO_4$+$NH_4NO_3$) was the highest (Fig. 7c). This suggests that nitrate played a more significant role than sulfate in affecting aerosol hygroscopic growth during the study period in Beijing.
[Lines 273-287]

**References**
Carrico, C. M., Rood, M. J., and Ogren, J. A.: Aerosol light scattering properties at Cape Grim, Tasmania, during the First Aerosol Characterization Experiment (ACE 1), J. Geophys. Res. Atmos., 103, 16,565–16,574, https://doi.org/10.1029/98JD00685, 1998.
Gysel, M., Crosier, J., Topping, D. O., Whitehead, J. D., Bower, K. N., Cubison, M. J., Williams, P. L., Flynn, M. J., McFiggans, G. B., and Coe, H.: Closure study between chemical composition and hygroscopic growth of aerosol particles during TORCH2, Atmos. Chem. Phys., 7, 6131-6144, https://doi.org/10.5194/acp-7-6131-2007, 2007.
Zhang, L.: Observation and model study of relative humidity effects on aerosol light scattering at regional backgound site in the Yangtze delta region, Ph.D. thesis, Chinese Academy of Meteorological Sciences, China, 107pp., 2017.
Zhang, L., Sun, J., Shen, X., Zhang, Y., Che, H. C., Ma, Q., Zhang, Y., Zhang, X., and Ogren, J. A.: Observations of relative humidity effects on aerosol light scattering in the Yangtze River Delta of China, Atmos. Chem. Phys., 15, 8439–8454, https://doi.org/10.5194/acp-15-8439-2015, 2015.
Zieger, P., Fierz-Schmidhauser, R., Gysel, M., Ström, J., Henne, S., Yttri, K. E., Baltensperger, U., and Weingartner, E.: Effects of relative humidity on aerosol light scattering in the Arctic, Atmos. Chem. Phys., 10, 3875–3890, https://doi.org/10.5194/acp-10-3875-2010, 2010.

Zieger, P., Fierz-Schmidhauser, R., Poulain, L., Müller, T. J., Birmili, W., Spindler, G., Wiedensohler, A., Baltensperger, U., and Weingartner, E.: Influence of water uptake on the aerosol particle light-scattering coefficients of the Central European aerosol, Tellus B., 66, 22716, http://dx.doi.org/10.3402/tellusb.v66.22716, 2014.

---

## Author Comment (AC2)

The effects of aerosol chemical composition on the relative humidity dependence of light scattering are presented for a site in Beijing. Parameterizations of f(RH) are developed for different observed conditions (e.g., very clean, moderately polluted, polluted based on measured light scattering levels). The paper is very well written and the figures (with one exception) clearly convey the results of the study. I only have minor comments – see below.

Line 39: change to "that REDUCES the amount". Also, please add a brief description of how SO2 control reduces the amount of sulfate.

**Reply:** Revised. L. Zhang et al. (2015) studied the relationship between the scattering enhancement factor and chemical composition in Lin'an, China, finding that nitrate has a stronger effect on aerosol hygroscopicity than sulfate has, partially due to the rigid control of $SO_2$ that reduces the amount of sulfate and increases the content of nitrite (Morgan et al., 2010). Apart from sea salt emissions and gypsum dust emissions during construction containing sulfate, sulfate is mainly formed by the oxidation of its gaseous precursor, SO2, in the atmosphere. In recent years, $SO_2$ emissions have been reduced substantially through a series of effective measures taken in China, like controlling the burning of loose coal and desulfurizing industrial equipment (Q. Zhang et al., 2019). Reducing $SO_2$ in the atmosphere thus directly affects the reduction in the sulfate content of aerosols. The saturated vapor pressure of nitric acid (HNO3) is higher than that of sulfuric acid (H2SO4), so the availability of ammonia (NH3) is key to the partitioning of HNO3. $HNO_3$ is often neutralized by $NH_3$ after $H_2SO_4$. Therefore, a reduction in $SO_2$ means that more $NH_3$ can be used to neutralize $HNO_3$, leading to higher nitrate concentrations, such as ammonium nitrate (HN4NO3), in aerosols (Monks et al., 2019).
[Lines 36-46]

Lines 109 – 111: Why is the absorption coefficient at 880 nm transformed into those at 525 nm? Doesn't the 7-wavelength aethalometer have a measurement wavelength near to 525 nm?

**Reply:** The absorption coefficient at 520 nm measured by the 7-wavelength aethalometer is more appropriate than using the absorption coefficient at 880 nm. So we chose absorption at 520 nm then converted it to 525 nm. The single-scattering albedo ($\omega_{0(525nm)}$) was also recalculated. Also updated were Figs. 2b, 2c, 3a and Table 1.
[Lines 130-132]

Lines 163 – 164: It is stated that "the proportion of organic matter and BC with weak hygroscopic abilities was low" from the southeast sector. Figure 3d indicates that mass fractions of BC were high in the southeast sector which seems to contradict this statement. Please clarify in the text.

**Reply:** The text has been revised as: "Figure 3c reveals that strongly hygroscopic aerosols with high values of $f$(RH = 85%, 525 nm) primarily came from the southeast sector. The proportion of secondary inorganics with strong hygroscopic abilities in aerosols from this direction was high, while the proportion of organic matter with weak hygroscopic abilities was low (Figs. 3e-f). Figure 3d indicates that the mass fraction of BC with weak hygroscopicity was slightly low in the southeast sector when wind speeds were lower than 4 m s$^{-1}$. However, when wind speeds were higher than 4 m s$^{-1}$, the mass fraction of BC was relatively high in this direction. Of all data associated with

southeast winds, identified were only three cases with wind speeds higher than 4 m s$^{-1}$, likely winds of short duration so not representative."

[Lines 180-186]

Figure 6: The inset figures showing organic mass fraction vs. f(RH) are difficult to read because of their size – especially if a reader is looking at a print version of the paper. I recommend putting the insets into a separate figure.

**Reply:** We have put the inset figures into a separate figure: Figure S6.

Lines 258 – 260: Please report the mass fractions of organics, SO4, and NO3 if they were provided in Malm et al. (2003), Pan et al. (2009), Quinn et al. (2005), and Yan et al. (2009). It is difficult to assess differences in the role of NO3 versus SO4 in determining f(RH) in these different regions without knowing the chemical composition reported in these previously published papers.

**Reply:** The proportions of organics, $SO_4^{2-}$, and $NO_3^-$ provided in Malm et al. (2003, 2005), Yan et al. (2009), Pan et al. (2009), and this study are listed in Table R1.

**Table R1: Proportions of sulfate, nitrate, and organic matter reported in previous studies and this study.**

| Site | Ammoniated sulfate or $SO_4^{2-}$/FM [%] | $NH_4NO_3$ or $NO_3^-$/FM [%] | OMC/FM [%] | Reference |
|------|------|------|------|------|
| BBNP | 51.0 | | 21.0 | Malm et al. (2003, 2005) |
| GC | 31 | 4.8 | 44 | |
| GSM | 63 | 0.8 | 25 | |
| BJ-1 | 28.0 | 16.0 | 34.0 | Yan et al. (2009) Sun et al. (2004) |
| BJ-2 | 23.0 | 12.0 | 36.0 | |
| BJ-3 | 23.0 | 14.0 | 29.0 | |
| XA#1 | 15.3 | 7.6 | 29.8 | |
| XA#2 | 17.7 | 9.4 | 24.4 | |
| XA#3 | 11.2 | 5.2 | 37.4 | |
| XA#4 | 5.3 | 1.5 | 40.0 | Pan et al. (2009) |
| XA#5 | 9.5 | 5.7 | 26.0 | |
| XA#6 | 6.1 | 1.2 | 36.7 | |
| XA#7 | 8.9 | 1.1 | 39.4 | |
| XA#8 | 10.8 | 3.2 | 43.0 | |
| BJ-CMA | 19.0 | 21.0 | 39.0 | This study |

Noet: BBNP was the observation site at the Big Bend National Park, Texas.

GC was the site at the Grand Canyon.

GSM was the site at the Great Smoky Mountains.

BJ-1, BJ-2, and BJ-3 were, respectively, affected by traffic emissions, industrial emissions, and anthropogenic emissions.

XA was the site at the Xin'An weather operational station in Baodi County. The symbols '#1' - '#8' represent different sampling dates.

FM: The FM at the XA site represents $PM_{2.1}$. The FMs at the other sites represent $PM_{2.5}$.

Lines 261 – 263: Does this mean the Chinese government has made more efforts to control SO2 emissions than other governments or has made more efforts to control SO2 than NOx emissions? Please clarify in the text.

**Reply:** To address this comment, the following has been added to the revised manuscript:

"In recent years, the Chinese government has made more efforts to control $SO_2$ emissions (Q. Zhang et al., 2019), e.g., adjusting and optimizing industrial capacities. Clean fuels have also been promoted in the residential sector, with trials for using clean energy in heating in northern China carried out in all "2+26" cities and in the Fenhe and Weihe River Plains. In addition, compliance with industrial emission standards has been strengthened. Desulfurization technology has been also applied to many heavy industrial facilities. However, China has many small-scale manufacturing enterprises, so it is much more difficult to regulate $NO_x$ emissions than $SO_2$ emissions. H. Li et al. (2109) have reported that emissions of $SO_2$ and $NO_x$ in 2017 dropped by 79.9% and 38.1%, respectively, from 2014 levels in Bejing, China. In 2020, $SO_2$ and primary $PM_{2.5}$ emissions dropped to one million tons, while $NO_x$ and volatile organic compound emissions were still ten million tons."

[Lines 287-295]

Line 315: Please define "DF".

**Reply:** Done. DF is the difference between $f$(RH = 85%, 525 nm) and $f$(RH = 80%, 525 nm), i.e., $f$(RH = 85%, 525 nm) minus $f$(RH = 80%, 525 nm).

[Lines 347-348]

**References**

Gysel, M., Crosier, J., Topping, D. O., Whitehead, J. D., Bower, K. N., Cubison, M. J., Williams, P. L., Flynn, M. J., McFiggans, G. B., and Coe, H.: Closure study between chemical composition and hygroscopic growth of aerosol particles during TORCH2, Atmos. Chem. Phys., 7, 6131–6144, https://doi.org/10.5194/acp-7-6131-2007, 2007.

Li, H., Cheng, J., Zhang, Q., Zheng, B., Zhang, Y., Zheng, G., and He, K.: Rapid transition in winter aerosol composition in Beijing from 2014 to 2017: response to clean air actions, Atmos. Chem. Phys., 19, 11,485–11,499, https://doi.org/10.5194/acp-19-11485-2019, 2019.

Malm, W. C., Day, D. E., Kreidenweis, S. M., Collett, J. L., and Lee, T.: Humidity-dependent optical properties of fine particles during the Big Bend Regional Aerosol and Visibility Observational Study, J. Geophys. Res. Atmos., 108, 4279, https://doi.org/10.1029/2002JD002998, 2003.

Malm, W. C., Day, D. E., Kreidenweis, S. M., Collett, J. L., Carrico, C., McMeeking, G., and Lee, T.: Hygroscopic properties of an organic-laden aerosol, Atmos. Environ., 39, 4969–4982, https://doi:10.1016/j.atmosenv.2005.05.014, 2005.

Monks, P. S., Granier, C., Fuzzi, S., Stohl, A., Williams, M. L., Akimoto, H., Amann, M., Baklanov, A., Baltensperger, U., Bey, I., Blake, N., Blake, R. S., Carslaw, K. S., Cooper, O. R., Dentener, F. J., Fowler, D., Fragkou, E., Frost, G. J., Generoso, S., Ginoux, P., Grewe, V., Guenther, A., Hansson, H. C., Henne, S., Hjorth, J., Hofzumahaus, A., Huntrieser, H., Isaksen, I. S. A., Jenkin, M. E., Kaiser, J., Kanakidou, M., Klimont, Z., Kulmala, M., Laj, P., Lawrence, M. G., Lee, J. D., Liousse, C., Maione, M., McFiggans, G. B., Metzger, A., Mieville, A., Moussiopoulos, N., Orlando, J. J., O'Dowd, C. D., Palmer, P. I., Parrish, D. D., Petzold, A., Platt, U., Poschl, U.,

A.S.H. Prévôt, A. S. H., Reeves, C. E., Reimann, S., Rudich, Y., Sellegri, K., Steinbrecher, R., Simpson, D., ten Brink, H., Theloke, J., van Der Werf, G. R., Vautard, R., Vestreng, V., Vlachokostas, C., and von Glasow, R.: Atmospheric composition change – global and regional air quality, Atmos. Environ., 43, 5268–5350, https://doi.org/10.1016/j.atmosenv.2009.08.021, 2009.

Morgan, W. T., Allan, J. D., Bower, K. N., Esselborn, M., Harris, B., Henzing, J. S., Highwood, E. J., Kiendlerscharr, A., Mcmeeking, G. R., Mensah, A. A., Northway, M. J., Osborne, S., Williams, P. I., and Krejci, R.: Enhancement of the aerosol direct radiative effect by semi-volatile aerosol components: airborne measurements in northwestern Europe, Atmos. Chem. Phys., 10, 8151–8171, https://doi.org/10.5194/acp-10-8151-2010, 2010.

Pan, X. L., Yan, P., Tang, J., Ma, J. Z., Wang, Z. F., and Gbaguidi, A.: Observational study of aerosol hygroscopic growth factors over rural area near Beijing megacity, Atmos. Chem. Phys., 9, 5087–5118, https://doi.org/10.5194/acpd-9-5087-2009, 2009.

Sun, Y., Wang, Z., Dong, H., Yang, T., Li, J., Pan, X., Chen, P., and Jayne, J. T.: Characterization of summer organic and inorganic aerosols in Beijing, China with an aerosol chemical speciation monitor, Atmos. Environ., 51, 250–259, https://doi.org/10.1016/j.atmosenv.2012.01.013, 2012.

Yan, P., Pan, X., Tang, J., Zhou, X., Zhang, R., and Zeng, L.: Hygroscopic growth of aerosol scattering coefficient: a comparative analysis between urban and suburban sites at winter in Beijing, Particuology, 7, 52–60, https://doi.org/10.1016/j.partic.2008.11.009, 2009.

Zhang, L., Sun, J., Shen, X., Zhang, Y., Che, H. C., Ma, Q., Zhang, Y., Zhang, X., and Ogren, J. A.: Observations of relative humidity effects on aerosol light scattering in the Yangtze River Delta of China, Atmos. Chem. Phys., 15, 8439–8454, https://doi.org/10.5194/acp-15-8439-2015, 2015.

Zhang, Q., Zheng, Y., Tong, D., Shao, M., Wang, S., Zhang, Y., Xu, X., Wang, J., He, H., Liu, W., Ding, Y., Lei, Y., Li, J., Wang, Z., Zhang, X., Wang, Y., Cheng, J., Liu, Y., Shi, Q., Yan, L., Geng, G., Hong, C., Li, M., Liu, F., Zheng, B., Cao, J., Ding, A., Gao, J., Fu, Q., Huo, J., Liu, B., Liu, Z., Yang, F., He, K., and Hao, J.: Drivers of improved $PM_{2.5}$ air quality in China from 2013 to 2017, P. Natl. Acad. Sci. USA, 116, 24,463–24,469, https://doi.org/10.1073/pnas.1907956116, 2019.

---

## Author Response (AR2)

We are very grateful to the numerous constructive suggestions and very detailed comments !

- In general, f(RH) is called the "particle light scattering enhancement factor" (or short scattering enhancement factor) and I would like to encourage you to use this term instead of "hygroscopic enhancement factor" which is too general. Hygroscopic growth in general can influence/increase or enhance the particle size, the light back scattering, and other parameters.

Reply: The terms have been corrected per your suggestion.

- In the abstract: I would suggest to re-phrase the sentence starting in line 18 to something like: "The effect is measured by the particle light scattering enhancement f(RH), where RH denotes the relative humidity, which is found to be …"

Reply: Revised.

- Line 31: Replace "ambient" with "elevated".

Reply: Revised.

- Line 32: Add "usually" before RH<40%. The 40% RH is a recommendation from WMO/GAW for aerosol monitoring and dependent on the instrument and set-up.

Reply: Done.

- Line 61: In our 2015-paper we actually added further measurements from Hyytiälä (Finland) and the correlations to sulfate and nitrate were actually much higher (see Table 3 and e.g. Figure 6 in https://acp.copernicus.org/articles/15/7247/2015/doi:10.5194/acp-15-7247-2015)

Reply: The pertinent finding of your paper is referred to in the revised manuscript: Zieger et al. (2015) also found that the mass fraction of sulfates was strongly corelated with $\gamma$, while the mass fraction of nitrates had a low correlation in Hyytiälä, Finland.

For the measurements at different sites, a common linear behavior has been found for inorganic and organic matter mass fractions, while individual inorganic matter such as nitrates or sulfates may show different functional dependencies (L. Zhang et al., 2015; Zieger et al., 2014, 2015). So more filed measurements including the $f$(RH) and chemical composition of aerosol in different seasons and regions should be added to obtain a more reliable estimate of $f$(RH).

- Section 2.1: More information on the actual inlet needs to be added here (Inlet with cut-off? Length? Was it heated? Drying applied? Average RH within the sampling line? What were the flows? etc).

Reply: We appreciate the suggestion. The instruments used in this field experiment include a dual-nephelometer system (Aurora 3000, Ecotech), an aerosol chemical speciation monitor (ACSM; Aerodyne Research Inc.) and a seven-wavelength aethalometer (AE33, Magee Scientific). They were all located in a mobility container on the ground. There are two air conditioners inside the container whose temperature was maintained at about 23℃. Sample air (16.7 lmp) went through a $PM_{2.5}$ cyclone inlet at about 4m above the ground, which only allowed particles with an aerodynamic diameter smaller than 2.5 μm to enter, and then dried by a Nafion dryer (MD-700-36F-3, Perma Pure LLC). The average RH within the sampling line was

about 30%. The sample air was not heated.

- Sect 2.2: Did you apply the truncation and illumination correction for your humidified nephelometer measurements?

Reply: In the paper, the truncation and illumination correction has been applied to the scattering coefficient with the Mie scattering calculation using the observational data of aerosol size distribution. However, there were some mistakes in the observed data of aerosol size distributions. So now, we use the AO98 method by Anderson and Orgen (1998) to correct the scattering coefficient again in the new revision, and all the relevant variables, such as scattering coefficient ($\sigma_{sp,525nm}$), single scattering albedo ($\omega_{0(525nm)}$) and scattering Ångström exponent ($\alpha_{(450nm-635nm)}$) are replaced with the newly corrected data in the current revision.

For the $f$(RH) calculations, there are no truncation and illumination correction applied to the scattering coefficients of both dry and humidified nephelometer. The comparison of the deviation between corrected and uncorrected $f$(RH=85%,525nm) is shown in Figure R1. The linear least square regression slop ± standard deviation is 1.064±0.002, the intercept ± standard deviation is -0.082±0.004 and $R$ is 0.999. The fitted line is very close to the line of 1:1. The uncorrected $f$(RH=85%,525nm) is a little lower than the corrected $f$(RH=85%,525nm).

[Figure]

**Figure R1: The relationship between the uncorrected $f$(RH=85%,525nm) and corrected $f$(RH=85%,525nm). The solid red line represents the linear least square regression. The blue line is the line of 1:1. The linear regression function and the Pearson's correlation coefficient ($R$) are given in the bottom-right corner of the panel.**

- Line 114: Please state the actually measured mean +/- STD of the sample RH here ("usually < 40%" is not sufficient).

Reply: Done. The actually measured mean ± standard deviation of the sample RH was 28.75±5.50%.

- Line 119: You state that calibrations were performed once a month but you only present 14 days of data. Please be more specific here by stating when which calibrations were performed.

Reply: Manual 'full calibration' and zero check and span check of the two nephelometers were performed at 10:30 on September 19th, 2019.

- I am still a bit puzzled (similar as reviewer #1) about the calculations of the dew point temperatures (Eq 1). The current equation 1 would need a proper reference. Why have you not used the Magnus formula?

Reply: Equation 1 was from the book "Hydrology: Water quantity and quality control" by Martin P. Wanielista, Robert Kersten and Ron Ealgin in 1997. James D. Johnston et al. (2015) also referenced this equation in their paper too.

To verify the accuracy of corrected results by equation 1, we used the Magnus formula (Eq. (R1)) to calculate the RH in the optical chamber of the wet nephelometer again.

$$e_w(t) = 611.2 \times e^{\frac{17.62t}{243.12+t}} \tag{R1}$$

The parameter t is the temperature in °C. The parameter $e_w$ is the saturation vapor pressure at t°C. The results of the calculated RH using these two methods are compared in Figure R2.

As shown in Fig. R2, the linear least square regression slop ± standard deviation is 0.997±0.00001 and the intercept ± standard deviation is 0.030±0.00086. The Pearson's correlation coefficient ($R$) between the RH$_{chamber}$ corrected by these two different methods equals 1. It illustrates that the calculated results using these two methods are highly consistent.

[Figure]

**Figure R2: The consistency of the RH$_{chamber}$ corrected by equation 1 (RH$_1$) and equation R1 (RH$_2$). The solid red line represents the linear least square regression. The blue line is the line of 1:1. The linear regression function and the Pearson's correlation coefficient ($R$) are given in the bottom-right corner of the panel.**

- Line 150 and SI (supplementary information): There is something wrong in your figure S8 (which should be referenced here as well): The values of the y-axis do not make sense and they are too low. Please check. Why is the humidogram of pure ammonium sulfate already increasing up to the deliquescence RH? Was the solution contaminated? Did you use mono-disperse aerosol? Please give more technical details on how the calibration was performed. I can recommend to have a look at the paper by Fierz-Schmidhauser et al (2010) who give details on the calibration of humidified nephelometers using known salts.

Reply: Thank you so much for noting the mistake we made in Fig. S4 (the original figure is Fig. S8), and we have replaced the plot with new one in the revised manuscript.

About the humidogram of pure ammonium sulfate increasing up to the deliquescence RH, we checked the calibration records, and found that the calibration result presented in the manuscript was wrong. Actually, we did twice calibrations of humidified nephelometers in September 21$^{st}$, 2019, only the first calibration was well performed. The schematic diagram of experimental set-up for calibration of dual-nephelometer system is illustrated in Figure R3. Ammonium sulfate aerosol was generated by nebulizing the aqueous ammonium sulfate solutions with a nebulizer (model SH600) and then dried through a drying tube without size selections. Then the dried ammonium sulfate aerosol (usually RH<5%) was carried and mixed with the filtered compressed pure air at flowrate of 5 lpm into the dual-nephelometer system.

[Figure]

**Figure R3: the schematic diagram of experimental set-up for the calibration of high-resolution humidified nephelometer system.**

Figure R4 shows the measured humidogram of $f$(RH,525nm) for ammonium sulphate, where x-axis represents the RH in the optical chamber of humidified nephelometer. When RH is lower than 79.41%, the values of $f$(RH,525nm) are consistently remained about 1. The literature value for the deliquescence relative humidity (DRH) of ammonium sulphate is 80% at 298K (Cheung et al., 2015). Figure R4 shows that the measured phase transition occurs at RH=80.07%. It illustrates that the RH inside the nephelometer chamber is correct and that the system is functioning properly.

[Figure]

**Figure R4: The $f$(RH) vs. RH of ammonium sulfate particles at λ=525nm. The phase transition at deliquescence occurred at about RH=80.07%.**

Figure R5 shows the result of the second calibration. However, the humidogram of pure ammonium sulfate is already increasing up to the DRH. It is probably because that we did not clean up the container completely

before reformulating the solution, the ammonium sulfate solution was contaminated.

[Figure]

**Figure R5: The *f*(RH) vs. RH of ammonium sulfate particles at λ=525nm. The phase transition at deliquescence occurred at about RH=80.37%.**

- Line 157: An aethalometer does not directly measure BC and it is common to use the acronym "eBC" or "EBC" for "equivalent black carbon" by convention.
Reply: Thank you for your suggestion. We have revised it.

- Equation 6 and Line 200-201: To calculate the organic mass fraction, do you now use the salts or the just the ions like sulfate, nitrate, etc? What do you mean with "among others"? As it reads now, you would count certain ions more than once.
Reply: We now just use the inorganic salts including $(NH_4)_2SO_4$, $NH_4HSO_4$ and $NH_4NO_3$ to calculate the organic mass fraction ($F_{org}$). The misleading sentence has been revised.

- Line 210: This should be more an if-statement, so "If the values of gamma<78% and gamma>82% are about the same, then eta will be close to 0.".
Reply: Revised.

- Line 406: Zeiger->Zieger
Reply: Corrected.

- Line 412: Add "partially" before "affected"
Reply: It is added now. Thanks.

- Line 448: The reference of Meier et al, 2009 is not correct here since this work is not about the particle light scattering enhancement f(RH). In my precious work, I have modelled (and observed) the effect of particle size on f(RH), see e.g. our paper in ACP from 2013: https://acp.copernicus.org/articles/13/10609/2013/ Please also have a look at the recommendation section for humidified nephelometer measurements.
Reply: We have added the citation of this paper which is indeed very useful.

- Line 528: Is the strong correlation with or without the green points? Is R now the coefficient of

determination or the Pearson correlation coefficient? Please clarify here and throughout the manuscript (incl. the figures with regression lines) which r-value you are showing.

Reply: The correlation without the green points ($R$=0.69, $R$: the Pearson's correlation coefficient) is stronger than that with the green points ($R$=0.62). They are clarified in the manuscript.

- Line 549: What threshold or exact criteria was applied to identify those points as "cases with large systematic errors"? Please clarify.

Reply: The threshold applied to identify those points as "cases with large systematic errors" was that the light-scattering coefficient ($\sigma_{sp,525nm}$) was lower than 20 Mm$^{-1}$.

- Line 552: Ny-Alesunf -> Ny-Alesund

Reply: Revised.

- Eq 13: The second part is not defined. What is the parameter "n" doing here? It is also not used later on. I guess this is an empirical equation that you introduce here? Or is there a reference for the "steepness index"?

Reply: Yes, $n$ is an empirical parameter introduced in Eq. (10), while $\tau$ is the ratio of the derivatives of $f'$ (80%) over $f'$ (60%) to characterize the change of curvature of $f$(RH) with RH (L. Zhang et al., 2015). From Eq. 10, we can derive the two derivatives and their ratio:

$$f'\ (80\%) = m \times n \times \left(\frac{4}{5}\right)^{(n-1)}$$

$$f'\ (60\%) = m \times n \times \left(\frac{3}{5}\right)^{(n-1)}$$

$$\tau = \frac{f'\ (80\%)}{f'\ (60\%)} = \left(\frac{4}{3}\right)^{(n-1)}$$

The parameter $n$, as defined in Eq. (10), dominates the magnitude of the scattering enhancement, which dictates the change in the curvature of the humidogram as described by $\tau$.

L. Zhang et al. (2015) firstly introduced the steepness index $\eta$ to quantitatively describe the growth pattern in their study. Because we have already used $\eta$ to represent the hysteresis index, $\tau$ is used to describe the curvature of $f$(RH) humidogram in this paper.

- Line 679: What do you mean by "The compensating effect of aerosols may be one of the main reasons for this phenomenon?" Which compensating effect?

Reply: We found that the positive correlation between $f$(RH=85%,525nm) and the inorganic matter mass fraction was very strong when inorganic matter mass fractions were smaller than 40%. However, when inorganic matter mass fractions were larger than 40%, the increasing tendency in $f$(RH=85%,525nm) as the proportion of inorganic matter increased slowly decreased. A similar phenomenon was also found for the negative correlation between organic matter and $f$(RH=85%,525nm).

The compensating effect between the size and chemical composition of aerosol may be one of the main reasons for this phenomenon. Many previous studies have found that for particles with strong hygroscopicity (e.g. NaCl), if their particle number size distribution tends towards large particle sizes, their hygroscopic growth ability may be similar to that of smaller particles with weak hygroscopicity (e.g. NH$_4$HSO$_4$; Zieger

et al., 2010, 2013; Y. Wang et al., 2017, 2018). In this study, it may be that when the proportion of inorganic matter was high, the inorganic matter of aerosol is mainly composed of relatively large particles.

- Data availability: Please use a proper data repository with a given DOI (e.g. zenodo.org or other data bases) with proper commented uploaded data. The current link leads to a website in Chinese which is difficult to read and use.

Reply: Thank you for your suggestion. We submitted the data to a data repository called the Science Data Bank with proper descriptions. The DOI of the dataset is : 10.11922/sciencedb.00785. Please notice that DOI does not work as the data has not been published yet. It will be valid automatically as soon as ScienceDB publishes the data. Before that you can view the data by the dataset private access link: http://www.scidb.cn/en/s/pzUjMjm.

Data description:

| Sheet name | Column ID | Explanation |
| --- | --- | --- |
| Sheet 1 | Column 1 | Year |
| | Column 2 | Month |
| | Column 3 | Day |
| | Column 4 | Hour |
| | Column 5 | Minute |
| | Column 6 | Second |
| | Column 7 | The corrected scattering coefficients measured by the dry nephelometer at 635nm wavelength |
| | Column 8 | The corrected scattering coefficients measured by the dry nephelometer at 525nm wavelength |
| | Column 9 | The corrected scattering coefficients measured by the dry nephelometer at 450nm wavelength |
| | Column 10 | Original scattering coefficients measured by the dry nephelometer at 635nm wavelength |
| | Column 11 | Original scattering coefficients measured by the dry nephelometer at 525nm wavelength |
| | Column 12 | Original scattering coefficients measured by the dry nephelometer at 450nm wavelength |
| | Column 13 | The temperature in the optical chamber of the dry nephelometer |
| | Column 14 | The relative humidity in the optical chamber of the dry nephelometer |
| | Column 15 | The pressure in the optical chamber of the dry nephelometer |
| | Column 16 | The corrected scattering coefficients measured by the wet nephelometer at 635nm wavelength |
| | Column 17 | The corrected scattering coefficients measured by the wet nephelometer at 525nm wavelength |
| | Column 18 | The corrected scattering coefficients measured by the wet nephelometer at 450nm wavelength |
| | Column 19 | Original scattering coefficients measured by the wet nephelometer at 635nm |

|  |  | wavelength |
|---|---|---|
|  | Column 20 | Original scattering coefficients measured by the wet nephelometer at 525nm wavelength |
|  | Column 21 | Original scattering coefficients measured by the wet nephelometer at 450nm wavelength |
|  | Column 22 | The temperature in the optical chamber of the wet nephelometer |
|  | Column 23 | The relative humidity in the optical chamber of the wet nephelometer |
|  | Column 24 | The pressure in the optical chamber of the wet nephelometer |
|  | Column 25 | The temperature measured by the sensor located at the inlet of the wet nephelometer |
|  | Column 26 | The relative humidity measured by the sensor located at the inlet of the wet nephelometer |
|  | Column 27 | The temperature measured by the sensor located at the outlet of the wet nephelometer |
|  | Column 28 | The relative humidity measured by the sensor located at the outlet of the wet nephelometer |
|  | Column 29 | The corrected relative humidity in the optical chamber of the wet nephelometer |
| Sheet 2 | Column 1 | Time information |
|  | Column 2 | Wind speed |
|  | Column 3 | Wind direction |
|  | Column 4 | Temperature |
|  | Column 5 | Relative humidity |
|  | Column 6 | The corrected scattering coefficients measured by the dry nephelometer at 525nm wavelength |
|  | Column 7 | The absorbing coefficients at 525nm wavelength retrieved from the a seven-wavelength aethalometer (AE33) |
|  | Column 8 | The single scattering albedo at 525nm |
|  | Column 9 | The scattering Ångström exponent calculated by the corrected scattering coefficients at 635nm and 525nm wavelengths |
| Sheet 3 | Column 1 | Time information |
|  | Column 2 | The mass concentration of organic matter |
|  | Column 3 | The mass concentration of nitrate ions |
|  | Column 4 | The mass concentration of sulfate ions |
|  | Column 5 | The mass concentration of ammonium ions |
|  | Column 6 | The mass concentration of chloride ions |
|  | Column 7 | The mass concentration of black carbon |

Sheet 1 is the data measured by a dual-nephelometer system.

Sheet 2 is the hourly average meteorological data and optical properties data.

Sheet 3 is the data of the chemical constitution of aerosol.

- Line 944 (caption of Fig 2): Replace "processes" with "occurrence"

Reply: Done.

- Figure 11: The font size (especially of the text within the graphs) is too small. As for all figures, you can always move less important graphs to the SI to save space.

Reply: Thank you for your suggestions. The font size within the figure 9 (the original figure is Fig. 11) has been revised as big as possible. And original figure 1, figure 4 and figure 12 were moved to the supplementary information already.

**Reference:**

Anderson, T. L. and Ogren, J. A.: Determining aerosol radiative properties using the TSI 3563 integrating nephelometer, Aerosol Sci. Tech., 29, 57–69, https://doi.org/10.1080/02786829808965551, 1998.

Cheung, H. H. Y., Yeung, M. C., Li, Y. J., Lee, B. P., and Chan, C. K.: Relative humidity-dependent TDMA measurements of ambient aerosols at the HKUST supersite in Hong Kong, China, Aerosol Sci. Tech., 49, 643–654, https://doi.org/10.1080/02786826.2015.1058482, 2015.

James, D. J., Brianna, M. M., Dennis, E., Scott, C. C., and Scott, A. B.: Comparison of Single-Point and Continuous Sampling Methods for Estimating Residential Indoor Temperature and Humidity, J. Occup. Environ. Hyg., 12, 785-794, https://doi.org/10.1080/15459624.2015.1047024, 2015.

Wang, Y., Zhang, F., Li, Z., Tan, H., Xu, H., Ren, J., Zhao, J., Du, W., and Sun, Y.: Enhanced hydrophobicity and volatility of submicron aerosols under severe emission control conditions in Beijing, Atmos. Chem. Phys., 17, 5239–5251, https://doi.org/10.5194/acp-17-5239-2017, 2017.

Wang, Y., Li, Z., Zhang, Y., Du, W., Zhang, F., Tan, H., Xu, H., Fan, T., Jin, X., Fan, X., Dong, Z., Wang, Q., and Sun, Y.: Characterization of aerosol hygroscopicity, mixing state, and CCN activity at a suburban site in the central North China Plain, Atmos. Chem. Phys., 18, 11,739–11,752, https://doi.org/10.5194/acp-18-11739-2018, 2018.

Wanielista, M., Kersten, R., and Eaglin, R.: Hydrology: Water Quantity and Quality Control, 2nd edition, John Wiley & Sons, New York, the United States, 1997.

Zhang, L., Sun, J., Shen, X., Zhang, Y., Che, H. C., Ma, Q., Zhang, Y., Zhang, X., and Ogren, J. A.: Observations of relative humidity effects on aerosol light scattering in the Yangtze River Delta of China, Atmos. Chem. Phys., 15, 8439–8454, https://doi.org/10.5194/acp-15-8439-2015, 2015.

Zieger, P., Aalto, P. P., Aaltonen, V., Äijälä, M., Backman, J., Hong, J., Komppula, M., Krejci, R., Laborde, M., Lampilahti, J., de Leeuw, G., Pfüller, A., Rosati, B., Tesche, M., Tunved, P., Väänänen, R., Petäjä, T.: Low hygroscopic scattering enhancement of boreal aerosol and the implications for a columnar optical closure study, Atmos. Chem. Phys., 15(13), 7247–7267, https://doi.org/10.5194/acp-15-7247-2015, 2015.

Zieger, P., Fierz-Schmidhauser, R., Gysel, M., Ström, J., Henne, S., Yttri, K. E., Baltensperger, U., and Weingartner, E.: Effects of relative humidity on aerosol light scattering in the Arctic, Atmos. Chem. Phys., 10, 3875–3890, https://doi.org/10.5194/acp-10-3875-2010, 2010.

Zieger, P., Fierz-Schmidhauser, R., Weingartner, E., Baltensperger, U.: Effects of relative humidity on aerosol light scattering: results from different European sites, Atmos. Chem. Phys., 13, 10609-10631,

https://doi.org/10.5194/acp-13-10609-2013, 2013.

Zieger, P., Fierz-Schmidhauser, R., Poulain, L., Müller, T. J., Birmili, W., Spindler, G., Wiedensohler, A., Baltensperger, U., and Weingartner, E.: Influence of water uptake on the aerosol particle light-scattering coefficients of the Central European aerosol, Tellus B., 66, 22716, https://doi.org/10.3402/tellusb.v66.22716, 2014.

---

## Author Response (AR3)

We are very grateful to the detailed comments !

In line 105 you now mention that you have used the truncation and illumination correction by Anderson and Ogren (1998). This correction is specifically for TSI nephelometers only (with different angles and wavelengths, etc.). Since you are using an Ecotech nephelometer, you need to use a different approach (see e.g. correction by Müller et al., https://amt.copernicus.org/articles/4/1291/2011/). I assume that your overall results won't change much but small changes to the numbers and fits will happen.

Reply: In this paper, we did use the approach applied by Müller et al. (2011). Because the approach applied by Müller et al. (2011) was based on the research by Anderson and Ogren (1998), we only quoted the latter paper, but we should have quoted both. The sentence in line 105 has been revised as "The truncation and illumination correction of the scattering coefficients has been done following Müller et al. (2011) that was developed specifically for Ecotech nephelometers originated from Anderson and Ogren (1998) for TSI nephelometers".

In addition, one minor details: In line 53 (or 51), you need to define gamma or write it in a more general way (e.g., "scattering enhancement") since it is not clear to the reader at this point what gamma is.

Reply: Thank you for your suggestion. We have revised "γ" as "scattering enhancement".

**Reference:**

Anderson, T. L. and Ogren, J. A.: Determining aerosol radiative properties using the TSI 3563 integrating nephelometer, Aerosol Sci. Tech., 29, 57–69, https://doi.org/10.1080/02786829808965551, 1998.

Müller, T., Laborde, M., Kassell, G., Wiedensohler, A.: Design and performance of a three-wavelength LED-based total scatter and backscatter integrating nephelometer, Atmos. Meas. Tech., 4, 1291–1303, https://doi.org/10.5194/amt-4-1291-2011, 2011.